# Towards Explaining Deep Neural Network Compression Through a Probabilistic Latent Space

## Abstract

Despite the impressive performance of deep neural networks (DNNs), their computational complexity and storage space consumption have led to the concept of network compression. While DNN compression techniques such as pruning and low-rank decomposition have been extensively studied, there has been insufficient attention paid to their theoretical explanation. In this paper, we propose a novel theoretical framework that leverages a probabilistic latent space of DNN weights and explains the optimal network sparsity by using the information-theoretic divergence measures. We introduce new *analogous projected patterns* (AP2) and *analogous-in-probability projected patterns* (AP3) notions for DNNs and prove that there exists a relationship between AP3/AP2 property of layers in the network and its performance. Further, we provide a theoretical analysis that explains the training process of the compressed network. The theoretical results are empirically validated through experiments conducted on standard pre-trained benchmarks, including AlexNet, ResNet50, and VGG16, using CIFAR10 and CIFAR100 datasets. Through our experiments, we highlight the relationship of AP3 and AP2 properties with fine-tuning pruned DNNs and sparsity levels.

## 1 Introduction

A wide variety of real-world problems are investigated by deep neural networks (DNNs) which enable powerful learning across a wide range of task learning. DNNs have provided remarkable performance in solving complex problems and this has led to significant growth of interest in the area of neural networks. However; the higher the performance, the more the number of parameters, resulting in complexity and space consumption. While DNNs pruning methods Yeom et al. (2021); Blalock et al. (2020); Ding et al. (2019a); Lee et al. (2019); Park et al. (2020); Ding et al. (2019b); Liu et al. (2021); You et al. (2019) compress networks and tackle computational complexity challenges, due to the lack of sufficient theoretical explanation of such methods, it's difficult to understand their behaviors. In this paper, we propose a novel framework that explains DNN pruning using probabilistic latent spaces of DNN weights and divergence measures. Our main idea is built upon mapping the weight matrix of the layers in both the original network and its corresponding sparse/pruned network to probability spaces from a distribution where the weight matrices are the parameter sets. To clarify our approach: consider a network $F^{(L)}$ with $L$-layers and its sparse version, denoted by $\widetilde{F}^{(L)}$ with parameters $\omega$ and $\widetilde{\omega} = m \odot \omega$, respectively, in which $m$ is a binary mask. Suppose that $f^{(l)}$, filters of the $l$-th layer of $F^{(L)}$ with $M_l$ neurons has weight matrix of $\omega^{(l)} \in \mathbb{R}^{M_{l-1} \times M_l}$ and $\widetilde{\omega}^{(l)}$ is the weight matrices of $\widetilde{f}^{(l)}$ which is filters of the $l$-th layer of $\widetilde{F}^{(L)}$ again with $M_l$ neurons. For clarity, we consistently treat weight matrices as high-dimensional vectors. If $\omega^{(l)} \in \mathbb{R}^{M_{l-1} \times M_l}$ represents the weight matrix of layer $l$, its vectorized version is denoted by $\omega^{(l)} \in \mathbb{R}^{(M_{l-1} \times M_l) \times 1}$, where $M_l$ is the number of neurons in layer $l$. Note that $\omega^{(l)}$ is referred to as a matrix whenever we use the term "matrix"; otherwise, it is considered as a vector.

In our approach, $\omega^{(l)}$ is projected, using a projection map $\mathcal{P} : \mathbb{R}^{(M_{l-1} \times M_l) \times 1} \mapsto \mathbb{R}^{(M_{l-1} \times M_l) \times 1}$, to a probability space $\mathcal{Z}$ with random variables $Z$ and with a distribution, denoted by $P_{\omega^{(l)}}$, which has $\omega^{(l)}$ as its parameter set. Similarly, a probability space $\widetilde{\mathcal{Z}}$ with latent variables $\widetilde{Z}$ and distribution $P_{\widetilde{\omega}^{(l)}}$ with $\widetilde{\omega}^{(l)}$ as its parameter set is assigned to $\widetilde{\omega}^{(l)}$ (we illustrate this in Fig. 1). We

show that by computing the Kullback-Leibler (KL) divergence between these two distributions, $KL(P_{\omega^{(l)}} || P_{\widetilde{\omega}^{(l)}})$, we explain the behavior of the pruned network and how magnitude-based pruning maintain the task learning performance. We show that as pruned network's performance convergences to optimal unpruned network performance, the KL divergence between $P_{\omega^{(l)}}$ and $P_{\widetilde{\omega}^{(l)}}$ for given layer $l$ convergences to zero. Note that what distinguishes our method from Bayesian inference is that the networks considered in this paper are non-Bayesian networks, meaning that there is no prior distribution associated with the network parameters. Hence, the distributions are not a distribution of network parameters but rather they are the distribution of defined latent spaces to clarify the network pruning methods.

*Why our approach is important?* Our new framework enables a deeper understanding of the complex interplay between network pruning and probabilistic distributions, where the weight tensors serve as their key parameters. Our approach effectively explains the sparsity of networks using latent spaces, shedding light on the interpretability of pruned models.

**Main Contributions:** The key contributions of this study are outlined as follows: (1) We introduce novel *projection patterns* between layers of DNN and their corresponding compressed counterparts. These patterns allow us to provide an upper bound on the difference between the performance of a network and its pruned version when the projection is applied; (2) We leverage *probabilistic latent spaces and employ divergence measures between distributions* to explain the sparsity of networks. We show the impact of the differences between two new probabilistic spaces, constructed based on the weights of an original network and its sparse network, on the difference between original and sparse models' performances; and (3) Through *extensive experiments* conducted on two well-known CIFAR10 and CIFAR100 datasets, we validate our proposed theoretical framework on multiple benchmarks and showcase how our approach effectively explains the sparsity of networks.

**Notations.** For clarity and ease of notation, we define all relevant symbols below. As mentioned earlier, we treat $\omega^{(l)}$ as a weight vector by default unless explicitly mentioned as a matrix using the term "matrix". Consider two neural networks, denoted as $F^{(L)}$ and $\widetilde{F}^{(L)}$ with $L$ layers and filters of their $l$-th layer, i.e., $f^{(l)}$ and $\widetilde{f}^{(l)}$. We use $\omega^{(l)}$ and $\widetilde{\omega}^{(l)}$ to represent the weight matrices of the $l$-th layer in $F^{(L)}$ and $\widetilde{F}^{(L)}$, respectively. Additionally, $\omega^*$ and $\widetilde{\omega}^*$ as the optimal weight tensors for $F^{(L)}$ and $\widetilde{F}^{(L)}$. The performance difference between these two networks is quantified by the function $D(F, \widetilde{F})$. To measure the dissimilarity between distributions, we use the Kullback–Leibler divergence denoted as $KL(.||.)$ though the intuition behind our theoretical explanations is based on any distance metrix. Finally, the terms $m$, $\sigma$, and $\mathbb{E}$ correspond to a binary mask, activation function, and expectation operation, respectively.

## 2 PROBLEM FORMULATION

Consider input $\mathbf{X} \in \mathcal{X}$ and target $Y \in \mathcal{Y}$, with realization space $\mathcal{X}$ and $\mathcal{Y}$, respectively. Suppose $(\mathbf{X}, Y)$ have joint distribution $D$. Training a neural network $F^{(L)} \in \mathcal{F}$ is performed by minimizing a loss function (empirical risk) that decreases with the correlation between the weighted combination of the networks and the label:

$$\mathbb{E}_{(\mathbf{X},Y)\sim D} \{ L(F^{(L)}(\mathbf{X}), Y) \} = -\mathbb{E}_{(\mathbf{X},Y)\sim D} \{ Y \cdot (b + \sum_{F \in \mathcal{F}} \omega_F \cdot F^{(L)}(\mathbf{X})) \}. \quad (1)$$

We remove offset $b$ without loss of generality. Define $\ell(\omega) := - \sum_{F \in \mathcal{F}} \omega_F \cdot F^{(L)}(\mathbf{X})$, $\omega \in \mathbb{R}^d$. Therefore the loss function in (1) becomes $\mathbb{E}_{(\mathbf{X},Y)\sim D} \{ Y \cdot \ell(\omega) \}$ with optimal weight solution $\omega^*$. Note $\widetilde{\omega}^* = m^* \odot \omega^*$ is optimal sparse weight where $m \in [0,1]^d$ is the binary mask matrix ($d$ is dimension). The optimal sparse weight is achieved by $(m^*, \omega^*) := \underset{\omega, m}{\arg\min} \mathbb{E}_{(\mathbf{X},Y)\sim D} \{ Y \cdot \ell(\omega) \}$.

**Definition 1** *Analogous Projected Patterns (AP2) - Consider the projected map function* $\mathcal{P} : \mathbb{R}^{(M_{l-1} \times M_l) \times 1} \mapsto \mathbb{R}^{(M_{l-1} \times M_l) \times 1}$. *We say filter* $f^{(l)}$ *and its sparse version* $\widetilde{f}^{(l)}$ *have AP2 if for small* $\epsilon^{(l)}$, $\left\| \mathcal{P}(f^{(l)}) - \mathcal{P}(\widetilde{f}^{(l)}) \right\|^2 \leq \epsilon^{(l)}$. *Equivalently, we say layer* $l$ *with weight* $\omega^{(l)}$ *and it's sparse version with sparse weight* $\widetilde{\omega}^{(l)}$ *have AP2 if for small* $\epsilon^{(l)}$, $\left\| \mathcal{P}(\omega^{(l)}) - \mathcal{P}(\widetilde{\omega}^{(l)}) \right\|^2 \leq \epsilon^{(l)}$ *or if* $\left\| (\omega^{(l)} - \widetilde{\omega}^{(l)}) \right\|^2 \leq \epsilon^{(l)}$ *when we use identity map as the projection function* $\mathcal{P}$.

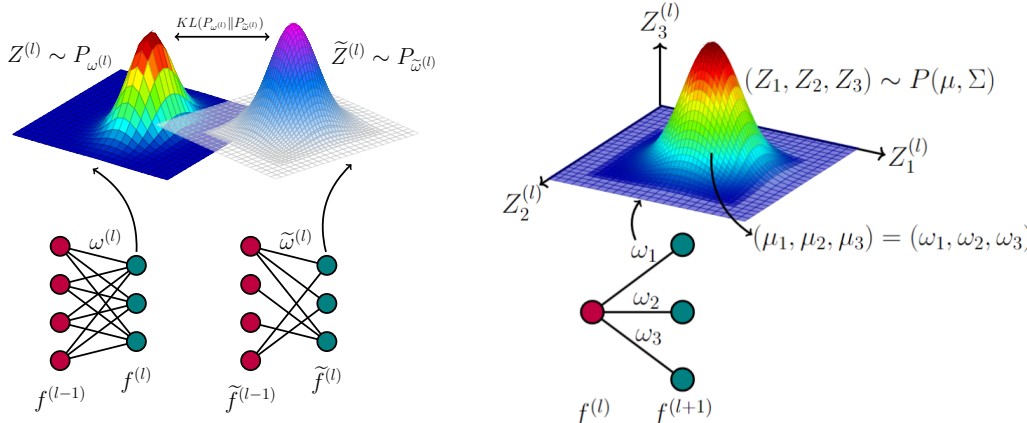

Figure 1: (Left) Proposed latent space $\mathcal{Z}$ and $\widetilde{\mathcal{Z}}$ from Gaussian distribution with mean parameters $\omega$ and $\widetilde{\omega}$ and covariances $\Sigma$ and $\widetilde{\Sigma}$, respectively: $P_{\omega^{(l)}} = G(\omega^{(l)}, \Sigma_1)$ and $P_{\widetilde{\omega}^{(l)}} = G(\widetilde{\omega}^{(l)}, \Sigma_2)$. (Right) A detailed illustration of maping weights matrices to the latent space as the distribution parameter.

Suppose latent variables $Z \in \mathcal{Z}$ and $\widetilde{Z} \in \widetilde{\mathcal{Z}}$ have distributions $P_\omega := P(., \omega)$ and $P_{\widetilde{\omega}} := P(., \widetilde{\omega})$ with parameter sets $\omega$ and $\widetilde{\omega}$, respectively. $\mathcal{Z}$ and $\widetilde{\mathcal{Z}}$ are latent spaces that could overlap. The Kullback-Leibler (KL) divergence Cover & Thomas (1991) between $P_\omega$ and $P_{\widetilde{\omega}}$ is

$$KL(P_\omega \| P_{\widetilde{\omega}}) := \mathbb{E}_{P_\omega} \left[ \log \left( P_\omega / P_{\widetilde{\omega}} \right) \right]. \tag{2}$$

**Definition 2** *Analogous-in-Probability Projected Patterns (AP3) - Consider the projection $\mathcal{P}$ that maps $\omega^{(l)}$, the l-th layer weight, to the probabilistic space $\mathcal{Z}$ with latent variable $Z$ and distribution $P_{\omega^{(l)}}$ with $\omega^{(l)}$ as its parameter set. Similarly projection $\mathcal{P}$ maps sparse layer l's weights, $\widetilde{\omega}^{(l)}$, to space $\widetilde{\mathcal{Z}}$ with latent variable $\widetilde{Z}$ and probability density $P_{\widetilde{\omega}^{(l)}}$. We say layer l and its sparse version have AP3 if for small $\epsilon^{(l)}$, $KL(P_{\omega^{(l)}} \| P_{\widetilde{\omega}^{(l)}}) \le \epsilon^{(l)}$.*

Throughout this paper, we assume that all distributions have positive definite covariance matrix. Note that the projection map in Definitions 1 and 2 are not necessarily the same. To clarify this, we provide an example that shows if layer $l$ and its sparse version have AP2 with an identity projection map, then they have AP3 property with Gaussian projection i.e. projection $\mathcal{P}$ maps $\omega^{(l)}$ and $\widetilde{\omega}^{(l)}$ to the latent space when latent variables $Z$ and $\widetilde{Z}$ have Gaussian distributions with means $\omega^{(l)}$ and $\widetilde{\omega}^{(l)}$, respectively. **Example - Gaussian Projection:** Suppose layer $l$ with weight $\omega^{(l)}$ and its sparse version with weight $\widetilde{\omega}^{(l)}$ have AP3 property with Gaussian projection i.e $P_{\omega^{(l)}} \sim G(\omega^{(l)}, \Sigma)$ and $P_{\widetilde{\omega}^{(l)}} \sim G(\widetilde{\omega}^{(l)}, \widetilde{\Sigma})$, where $\Sigma$ and $\widetilde{\Sigma}$ are covariance matrices. Using (2) we can write

$$KL(P_{\omega^{(l)}} \| P_{\widetilde{\omega}^{(l)}}) = \frac{1}{2} \Big( \log \frac{|\Sigma|}{|\widetilde{\Sigma}|} + tr(\Sigma^{-1}\widetilde{\Sigma}) + (\omega^{(l)} - \widetilde{\omega}^{(l)})^T \Sigma^{-1} (\omega^{(l)} - \widetilde{\omega}^{(l)}) - d \Big), \tag{3}$$

where $tr$ is the trace of the matrix, $d$ is the dimension and $\log$ is taken to base $e$. Without loss of generality assume that $\Sigma = \widetilde{\Sigma} = \Sigma$, then for constant $C$, (3) is simplified and bounded as $\frac{1}{2} \left( C + (\omega^{(l)} - \widetilde{\omega}^{(l)})^T \Sigma^{-1} (\omega^{(l)} - \widetilde{\omega}^{(l)}) \right) \le \frac{1}{2} \left( C + \lambda_{max} \|\omega^{(l)} - \widetilde{\omega}^{(l)}\|_2^2 \right)$. Here $\lambda_{max}$ is the maximum eigenvalue of matrix $\Sigma^{-1}$. Since layer $l$ and its sparse have AP2 property (identity projection), $\|\omega^{(l)} - \widetilde{\omega}^{(l)}\|_2^2 \le \epsilon^{(l)}$. This implies that layer $l$ and its sparse have AP3 property

$$KL(P_{\omega^{(l)}} \| P_{\widetilde{\omega}^{(l)}}) \le \frac{1}{2} \left( C + \lambda_{max} \epsilon^{(l)} \right), \quad \text{where } C = \log \frac{|\Sigma|}{|\widetilde{\Sigma}|} + tr(\Sigma^{-1}\widetilde{\Sigma}) - d \tag{4}$$

Note that by considering the assumption of the equality of the covariance matrices, $C = 0$, and for smaller $\epsilon^{(l)}$ the bound in (4) becomes tighter.
As observed in the previous example, it was demonstrated that for Gaussian distributions, the AP2 property implies the AP3 property. Building upon this finding, under the same distributional assumptions, based on Eq. 5, we infer that if the latent Gaussian distribution has a covariance matrix

$\Sigma$ in which all eigenvalues are greater than or equal to 1, then AP3 property holds true and also implies AP2. Note that $\lambda_{min}$ is the minimum eigenvalues of $\Sigma^{-1}$.

$$\|\omega^{(l)} - \widetilde{\omega}^{(l)}\|_2^2 \leq \lambda_{min}\|\omega^{(l)} - \widetilde{\omega}^{(l)}\|_2^2 \leq (\omega^{(l)} - \widetilde{\omega}^{(l)})^T \Sigma^{-1}(\omega^{(l)} - \widetilde{\omega}^{(l)})) \tag{5}$$

This means that the AP3 property implies the AP2 property for certain distributions. However, establishing this behavior theoretically for complex multivariate distributions is challenging due to the lack of exact KL values. Consequently, empirical evidence and numerical approximations are often used to study the AP3 property of such distributions.

**Definition 3** Csiszár & Körner (2011) (Pinsker's inequality) If $P$ and $Q$ are two distributions on a measurable space $(X, \mathcal{F})$, then $KL(P\|Q) \geq 2d_{TV}^2$.

**Definition 4** Cover & Thomas (1991) Consider a measurable space $(\Omega, \mathcal{A})$ and probability measures $P$ and $Q$ defined on $(\Omega, \mathcal{A})$ on a finite domain $D$. The total variation distance between $P$ and $Q$ is defined as

$d_{TV}(P, Q) := \frac{1}{2} \int |P - Q| \, dx = \sup_{A \in \mathcal{A}} |P(A) - Q(A)|.$

**Lemma 1** Nishiyama (2022) Let $P$ and $Q$ be two arbitrary distributions in $\mathbb{R}^d$ with mean vectors of $m_P$ and $m_Q$ and covariance matrices of $\Sigma_P$ and $\Sigma_Q$. Then, by setting constant $a := m_P - m_Q$,

$$d_{TV}(P, Q) \geq \frac{a^T a}{2(tr(\Sigma_P) + tr(\Sigma_Q)) + a^T a} \quad \text{where } tr(A) \text{ is the trace of a matrix } A.$$

**Corollary 1** *Let $P$ and $Q$ be two arbitrary distributions in $\mathbb{R}^d$ with mean vectors of $m_P$ and $m_Q$ and covariance matrices of $\Sigma_P$ and $\Sigma_Q$, respectively. Set constant $a := m_p - m_Q$. Using definition 3 and Lemma 1, $KL(P\|Q) \geq 2d_{TV}^2(P, Q) \geq 2\left(\frac{a^T a}{2(tr(\Sigma_P)+tr(\Sigma_Q))+a^T a}\right)^2$.*

According to the Corollary 1, for any two continuous distributions $P_{\omega^{(l)}}$ and $P_{\widetilde{\omega}^{(l)}}$ in a $d$-dimensional space, both with identity matrix $I \in \mathbb{R}^{d \times d}$ as their covariance matrices, if AP3 is satisfied, then, AP2 holds when $\epsilon < 2$. However, it's important to note that there can be instances where AP3 doesn't necessarily imply AP2, and this relationship depends on the specific values of $\epsilon$ and the characteristics of the distributions.

**Definition 5** *Approximately AP3 in Iteration $t$ - Consider the projection $\mathcal{P}$ that maps optimal weight $\omega^*$, to the probabilistic space $\mathcal{Z}$ with latent variable $Z$ and distribution $P_{\omega^*}$ with $\omega^*$ as its parameter set . Similarly projection $\mathcal{P}$ maps sparse networks optimal weights, $\widetilde{\omega}^*$, to space $\widetilde{\mathcal{Z}}$ with latent variable $\widetilde{Z}$ and distribution $P_{\widetilde{\omega}^*}$. Let $\widetilde{\omega}^t$ be the weight of pruned network $\widetilde{F}_t^{(L)}$ at iteration $t$ in training (with mapped latent space $\widetilde{\mathcal{Z}}$ from distribution $P_{\widetilde{\omega}^t}$). We say networks $\widetilde{F}_t^{(L)}$ and $F^{(L)}$ have approximately AP3 if for small $\epsilon_t$:*

$$|KL(P_{\omega^*}\|P_{\widetilde{\omega}^*}) - KL(P_{\omega^*}\|P_{\widetilde{\omega}^t})| = O(\epsilon_t), \quad \text{where as } \epsilon_t \to 0 \text{ when } t \to \infty. \tag{6}$$

## 3 THEORETICAL ANALYSIS

In DNNs compression, it is essential to maintain the performance with a core mission of reducing computational complexity and memory usage. However, in many compression methods, after significant computational improvement, we want the performance of the network after compression to be as close as possible to the best performance of the complete (non-compressed) network.

**Definition 6** *Performance Difference (PD) - Let $F^{(L)}$ and $\widetilde{F}^{(L)}$ be $L$ layers network and its corresponding compressed network. Using loss function in (1), we define PD by*

$$D(F^{(L)}, \widetilde{F}^{(L)}) := \left| \mathbb{E}_{(\mathbf{X},Y)\sim D}\left\{ L(F^{(L)}(\mathbf{X}), Y) \right\} - \mathbb{E}_{(\mathbf{X},Y)\sim D}\left\{ L(\widetilde{F}^{(L)}(\mathbf{X}), Y) \right\} \right|. \tag{7}$$

We are now in the position to state our first main finding and all proofs in details are provided in Appendix. Prior to introducing the theorems, it's crucial to underline that each latent space is presupposed to include a covariance matrix among its inherent parameters.

### 3.1 Performance Difference Bounds

**Theorem 1** *Let $\widetilde{\omega}^*$ be the optimal weight of trained sparse network $\widetilde{F}^{(L)}$. Let $P_{\omega^*}$ be distribution with $\omega^*$ as its parameters for trained network $F^{(L)}$ and let $P_{\widetilde{\omega}^*}$ be distribution with $\widetilde{\omega}^* = m^* \odot \omega^*$ as its parameters for trained sparse network $\widetilde{F}^{(L)}$. If there exists a layer $l \in \{1, \ldots, L\}$ that has AP3 with the corresponding layer in $\widetilde{F}^{(L)}$ i.e. $KL(P_{\omega^{*(l)}} \| P_{\widetilde{\omega}^{*(l)}}) \leq \epsilon^{(l)}$, for $0 < \epsilon^{(l)} < 2$, then the performance difference in (7) is bounded by $g(\epsilon^{(l)})$ that is*

$$D(F^{(L)}, \widetilde{F}^{(L)}) \leq g(\epsilon^{(l)}), \tag{8}$$

*where $g$ is an increasing function of $\epsilon^{(l)}$ i.e. for smaller $\epsilon^{(l)}$, the bound in (8) becomes tighter. Note that, trained sparse network reffers to the pruned network that has undergone fine-tuning after the pruning process.*

Theorem 1 is the immediate result of two Lemmas 2 and 3 below together. It is important to note that the function $g(\epsilon^{(l)})$ depends on $\epsilon^{(l)}$ sequentially through $\epsilon^{(L)}, \ldots, \epsilon^{(l-1)}$. This means that as in Lemma 3, $\epsilon^{(l+1)}$ is an increasing function of $\epsilon^{(l)}$ for $l, \ldots, L-1$ and according to Lemma 2 the function $g$ is an increasing function of $\epsilon^{(L)}$.
In Lemma 2 and Corollary 2, $\lambda_{\max}$ refers to the maximum eigenvalue of Hessian $\nabla^2 \ell(\widetilde{\omega}^*)$. Meanwhile, in Corollary 3, $\lambda_{\max}$ represents the maximum eigenvalue of the inverse covariance matrix $\Sigma^{-1}$. In both corollaries, $\lambda_{\min}$ denotes the minimum eigenvalue of $\Sigma^{-1}$. We also introduce constants $\epsilon^{(L)}$ and $\epsilon^{(l)}$ for reference.

**Lemma 2** *Consider a trained network $F^{(L)}$ and its trained sparse version, $\widetilde{F}^{(L)}$ with optimal weights $\omega^*$ and $\widetilde{\omega}^*$, respectively. Apply the projection map $\mathcal{P}$ on the last layer and let $P_{\omega^{*(L)}}$ and $P_{\widetilde{\omega}^{*(L)}}$ be distributions with $\omega^{*(L)}$ and $\widetilde{\omega}^{*(L)} = m^{*(L)} \odot \omega^{*(L)}$ as their parameters for $F^{(L)}$ and $\widetilde{F}^{(L)}$, respectively. Note that in $F^{(L)}$ and $\widetilde{F}^{(L)}$, layers $1, \ldots, L-1$ are not necessaraly identical. Assume $\text{Vol}(\mathcal{Z}^{(L)})$ is finite and the last layer of $F^{(L)}$ and $\widetilde{F}^{(L)}$ have AP3 property for $0 < \epsilon^{(L)} < 2$ i.e. $KL(P_{\omega^{*(L)}} \| P_{\widetilde{\omega}^{*(L)}}) \leq \epsilon^{(L)}$. Hence, the PD in (8) is bounded by $g^{(L)}(\epsilon^{(L)})$ i.e. $D(F^{(L)}, \widetilde{F}^{(L)}) \leq g^{(L)}(\epsilon^{(L)})$, where $\widetilde{F}^{(L)}$ is the sparse network with last layer weights $\widetilde{\omega}^{*(L)}$.*
*Here for constant $C$, $g^{(L)}(\epsilon^{(L)}) := \mathbb{E}_{(\mathbf{X},Y)\sim D}\left\{Y \cdot \left(\frac{\lambda_{max}\sqrt{\frac{\epsilon^{(L)}}{2}}C}{2(1-\sqrt{\frac{\epsilon^{(L)}}{2}})}\right)\right\}$, is an increasing function of $\epsilon^{(L)}$ i.e. for smaller $\epsilon^{(L)}$, the bound for $D(F^{(L)}, \widetilde{F}^{(L)})$ becomes tighter.*

**Proof sketch of Lemma 2:** While a detailed proof of Lemma 2 is available in Appendix, here's a concise overview. By considering Equation 1 and by applying Jensen inequality, we have $D(F^{(L)}, \widetilde{F}^{(L)}) \leq \mathbb{E}_{(\mathbf{X},Y)\sim D}\{Y \cdot |\ell(\omega^*) - \ell(\widetilde{\omega}^*)|\}$. Using both quadratic Taylor approximation for $\ell$ and Corollary 1, $D(F^{(L)}, \widetilde{F}^{(L)}) \leq \mathbb{E}_{(\mathbf{X},Y)\sim D}\left\{Y \cdot \left(\frac{\lambda_{max}\sqrt{\frac{\epsilon^{(L)}}{2}}C}{2(1-\sqrt{\frac{\epsilon^{(L)}}{2}})}\right)\right\}$, where $C$ is a constant.

**Corollary 2** *In Lemma 2, suppose the projection map function is Gaussian projection i.e. $P_{\omega^{*(L)}} \sim G(\omega^{*(L)}, \Sigma)$ and $P_{\widetilde{\omega}^{*(L)}} \sim G(\widetilde{\omega}^{*(L)}, \Sigma)$ with same positive definite covariance matrices. If the last layer of $F^{(L)}$ and $\widetilde{F}^{(L)}$ have AP3 with bound $\epsilon^{(L)} \geq 0$, then*

$$D(F^{(L)}, \widetilde{F}^{(L)}) \leq g_G^{(L)}(\epsilon^{(L)}), \quad \text{where } g_G^{(L)}(\epsilon^{(L)}) = \mathbb{E}_{(\mathbf{X},Y)\sim D}\left\{Y \cdot \left(\frac{\lambda_{max}}{2}\sqrt{\frac{2\epsilon^{(L)}}{\lambda_{min}}}\right)\right\}. \tag{9}$$

**Lemma 3** *For trained layer $l$ and sparse $l$-layer, let $P_{\omega^{*(l)}}$ and $P_{\widetilde{\omega}^{*(l)}}$ be distributions with $\omega^{*(l)}$ and $\widetilde{\omega}^{*(l)}$ as their parameters, respectively. Under the assumptions $KL(P_{\omega^{*(l)}} \| P_{\widetilde{\omega}^{*(l)}}) \leq \epsilon^{(l)}$ and $\text{Vol}(\mathcal{Z}) < \infty$, for $\epsilon^{(l)} \geq 0$, and $\mathbb{E}[1/(P_{\widetilde{\omega}^{*(l+1)}})^2]$ being bounded, we have $KL(P_{\omega^{*(l+1)}} \| P_{\widetilde{\omega}^{*(l+1)}}) \leq \epsilon^{(l+1)}$, where $\epsilon^{(l+1)} := C_{l+1}\left(\frac{\sqrt{2\epsilon^{(l)}}}{C_l \text{Vol}(\mathcal{Z})} + C_{\sigma^{-1},x}^{(l)}\right)$ for constants $C_l$, $C_{l+1}$, $C_{\sigma^{-1},x}^{(l)}$ and $\epsilon^{(l+1)}$ is increasing in $\epsilon^{(l)}$.*

**Corollary 3** *In Lemma 3, suppose the projection map function is Gaussian projection i.e. $P_{\omega^{*(l)}} \sim G(\omega^{*(l)}, \Sigma)$ and $P_{\widetilde{\omega}^{*(l)}} \sim G(\widetilde{\omega}^{*(l)}, \Sigma)$ with covariance matrix $\Sigma$ for $l = 1, \ldots, L-1$. If for $\epsilon^{(l)} \geq 0$, $KL(P_{\omega^{*(l)}} \| P_{\widetilde{\omega}^{*(l)}}) \leq \epsilon^{(l)}$ and $\text{Vol}(\mathcal{Z}) < \infty$, then for constants $C_1$, $C_{\sigma^{-1},\sigma,x}$, and $K$,*

$$KL(P_{\omega^{*(l+1)}} \| P_{\widetilde{\omega}^{*(l+1)}}) \leq \epsilon^{(l+1)}, \quad \text{where } \epsilon^{(l+1)} := \frac{1}{2}\left(C_1 + \lambda_{max}C_{\sigma^{-1},\sigma,x}^2\left(\frac{2\epsilon^{(l)} - K}{\lambda_{min}}\right)\right). \tag{10}$$

### 3.2 How AP3 Explains Training Sparse Network?

Theorem 1 links the AP3 property to the performance difference between Networks $F^{(L)}$ and $\widetilde{F}^{(L)}$ on optimal weights $\omega^*$ and $\widetilde{\omega}^*$. A follow-up question is *how the latent space $\mathcal{Z}$ explains the training convergence of compressed network $\widetilde{F}^{(L)}$ to optimal sparsity level with approximately maintaining the performance*. To provide an answer to this question we employ Definition 5 and approximately AP3 in iteration $t$ property as stated in the next theorem.

**Theorem 2** *Suppose that $P_{\omega^*}$ is the distribution of latent variable $Z \in \mathcal{Z}$ for network $F^{(L)}$ and $P_{\widetilde{\omega}^*}$ is the distribution of $\widetilde{Z} \in \widetilde{\mathcal{Z}}$ for sparse network $\widetilde{F}^{(L)}$. Let $\widetilde{\omega}^t$ be the weight of pruned network $\widetilde{F}_t^{(L)}$ after iteration $t$ of training with total $T$ iterations. Under the assumption that $F^{(L)}$ and $\widetilde{F}_t^{(L)}$ have approximately AP3 in iteration $t$ i.e. for small $\epsilon_t$, $|KL(P_{\omega^*}\|P_{\widetilde{\omega}^*}) - KL(P_{\widetilde{\omega}^*}\|P_{\widetilde{\omega}^t})| = O(\epsilon_t)$, then we have $D(F^{(L)}, \widetilde{F}_t^{(L)}) = O(\epsilon_t, T)$, equivalently $\left|\mathbb{E}_{(\mathbf{X},Y)\sim D}\left[Y.\ell(\omega^*)\right] - \mathbb{E}_{(\mathbf{X},Y)\sim D}\left[Y.\ell(\widetilde{\omega}^t)\right]\right| = O(\epsilon_t, T)$.*

## 4 Related Work

Pruning has been a widely-used approach for reducing the size of a neural network by eliminating either unimportant weights or neurons. Existing methods can be categorized into structured and unstructured pruning. structured pruning, studied in Ding et al. (2019a;b); Liu et al. (2021); You et al. (2019); Anwar et al. (2017); Wen et al. (2016); Li et al. (2016); He et al. (2017); Luo et al. (2017), involves removing entire structures or components of a network, such as layers, channels, or filters. However, unstructured pruning aims to reduce the number of parameters by zeroing out some weights in the network. Some popular techniques commonly used in unstructured pruning are weight magnitude-based pruning, L1 regularization, Taylor expansion-based pruning, and Reinforcement learning-based pruning Han et al. (2015); Lee et al. (2019); Park et al. (2020); Sanh et al. (2020); Molchanov et al. (2016); Zhu & Gupta (2017); Ye et al. (2018).
In this paper, to provide a comprehensive explanation of our findings, we focus on weight magnitude pruning technique, previously examined in Han et al. (2015); Molchanov et al. (2016; 2019). In Han et al. (2015), where weight magnitude pruning is introduced, the authors suggested that a substantial number of network weights can be pruned without causing significant performance deterioration. To establish this notion, they conducted experiments employing models such as AlexNet and VGG16. While their research centers on empirically showcasing the efficacy of such technique, it doesn't extensively explore the formal theoretical analysis of the subject. Our intent, instead, is to provide a theoretical foundation using probability latent spaces to elucidate the principles of network pruning, particularly weight pruning, and its significant role in preserving accuracy.

## 5 Experiments

To validate our theoretical analysis empirically, contributing to a more comprehensive understanding of the sparsification techniques in DNNs, we conduct a set of experiments on two common CIFAR10 Krizhevsky et al. (2009) and CIFAR100 Krizhevsky et al. (2009) datasets, using three different pre-trained benchmark models: ResNet50 He et al. (2016) and VGG16 Simonyan & Zisserman (2014), which are pre-trained on ImageNet. The results on AlexNet Krizhevsky et al. (2012) are provided in Appendix.

### 5.1 Setup:

*Training Phase:* We initialized these models' weights with pre-trained weights as the baseline models. To ensure the results' significance we performed three trials of training the models. Each trial involved training for 100 epochs using the SGD optimizer with a momentum of 0.9 and learning rates of 0.1 for pre-trained ResNet50 and 0.01 for AlexNet and VGG16. In order to facilitate convergence and enhance training efficiency, we incorporated a learning rate scheduler that dynamically adjusted the learning rate as the model approached convergence.
*Pruning Phase:* We investigate the effects of weight differences within the last layer of the baseline and its pruned version, using magnitude-based pruning Han et al. (2015), over the PD. We initialize the model with the best parameters obtained from each trial. Three different pruning methods

were employed: lowest (removing weights with the lowest magnitudes targeting connections that contributed the least to the model's performance), highest (removing weights with the highest magnitudes, thus eliminating the most influential connections), and random (a stochastic element by randomly removing weights) magnitude prunings. Different percentages of sparsity level, including 0.1, 0.3, 0.5, and 0.8, are used to provide empirical evidence to support the validity of Lemma 2 for the optimal pruning percentage. We maintained similar experimental setups as before. However, given that we were fine-tuning the trained parameters after pruning, we limited the training to 20 epochs. Note that during each epoch of fine-tuning, the test accuracy of the pruned model is computed for the purpose of comparison with the test accuracy of the original network.

The experimental results section is divided into three main segments, all centered around explaining optimal network sparsity using information-theoretic divergence measures: Section 5.2 investigates the relationship between AP2 and the performance difference of the original network and its sparse version by focusing on the last layer of the network, known as Lemma 2; Section 5.3 extends our analysis to AP3 by considering multivariate T-Student distribution; Finally, section 5.4 involves a crucial comparison between AP2 and AP3, aimed at determining whether AP2 truly constitutes the optimal explanation for PDs or if an opportunity exists to identify parameter sets within distributions that might surpass AP2's capabilities.

## 5.2    Multivariate Gaussian Distribution with Diagonal Covariance Matrices

In this section, we aim to validate Lemma 2 by examining the projections of the last layer weights of the networks and their pruned version onto multivariate Gaussian distributions. Specifically, we analyze the mean values of $\omega^{(L)}$ and $\widetilde{\omega}^{(L)}$ in their respective latent spaces, $\mathcal{Z}$ and $\widetilde{\mathcal{Z}}$. We utilize a coefficient of the identity covariance matrix for both latent spaces. By substituting this covariance matrix into the Gaussian distribution example outlined in Section 2, we observe that the KL-divergence value precisely matches AP2 (Gaussian Projection Example). Consequently, the following experiments provide strong evidence for the correctness of Lemma 2 for AP2 across various pre-trained models. *Note that in Figs. 2 and 3 Y-axis shows the value of both AP2 and PD computed on testset.*
**ResNet50:** Fig. 2 shows a compelling visual demonstration, reaffirming the correctness of Lemma 2 through the use of AP2 on the pruned ResNet50 model applied to CIFAR10 and CIFAR100 datasets. It is crucial to emphasize that in Fig. 2, while the highest and random magnitude pruning methods consistently validate Lemma 2, the lowest magnitude pruning presents occasional challenges. Specifically, in cases where we prune $30\%$ and $50\%$ of the weights, the performance difference of these two pruning percentages is less than $10\%$. These discrepancies can be attributed to significant differences between theoretical assumptions and experimental observations. For instance, one crucial theoretical assumption in Lemma 2 is the use of optimal weights.
Another factor, especially in lowest magnitude pruning, is the removal of non-essential weights, minimally impacting both AP2 and performance differences. Consequently, this can lead to overlapping results, particularly when stochastic variations come into play. However, despite these occasional overlaps, the close values still provide valuable insights into understanding of optimal network sparsity using information-theoretic divergence measures of the latent distributions.
Further, we conducted another experiment by comparing specific percentages of different pruning methods simultaneously, which is shown in the second row of Fig 2. Despite some variability due to stochastic gradient descent in certain epochs, the expected behavior described in Lemma 2 becomes evident. This signifies that when one pruning method's KL-divergence decreases relative to another, its PD similarly diminishes. Note that covariance matrices for all methods and different percentages are the same in order to compare them in fairness.
**VGG16:** To explore the interpretability of network sparsity using the AP2 metric extensively we investigated VGG16 additionally in Fig 3.

We ran additional experiments using Gaussian distribution with non-diagonal covariance matrices, which their KL-divergence serves as AP3. Detailed visual representations are included in Appendix to support our research findings.

## 5.3    Multivariate T-Student Distribution

Here, we employed the multivariate T-Student distribution, suited for heavy-tailed distributions and outliers in the latent space, enhancing the informativeness and interpretability of AP3 over AP2. We use the Monte Carlo technique to estimate KL-divergence in section 5.2. We set the degrees

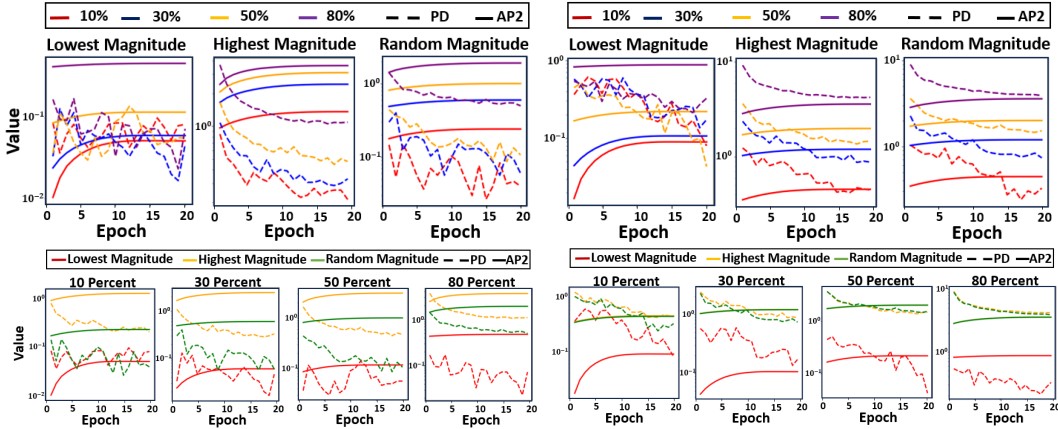

Figure 2: Comparing AP2 & PD for pruned ResNet50. Top- Comparing four pruning percentages for a given pruning method. Bottom-Comparing three pruning methods for a given pruning %. Expriments are done on CIFAR10 (Left) and CIFAR100 (Right).

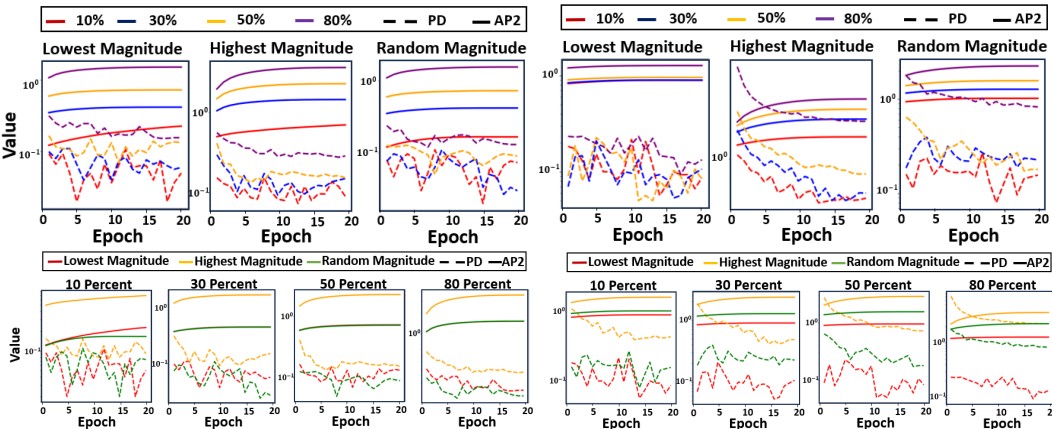

Figure 3: Comparing AP2 & PD for pruned VGG16. Top- Comparing four pruning percentages for a given pruning method. Bottom-Comparing three pruning methods for a given pruning %. Expriments are done on CIFAR10 (Left) and CIFAR100 (Right).

of freedom in the T-student distribution to $4$, aiming for heavier-tailed distributions and greater dissimilarity from a Gaussian distribution and generated 600,000 data points for improved estimation accuracy. To efficiently estimate KL divergence in high-dimensional spaces, we grouped weight vectors into 100 groups and averaged them, resulting in a 100-dimensional latent variable for KL-divergence estimation. While results for RexNet50 are presented here, experiments on AlexNet and VGG16 are detailed in Appendix. In Fig 4 (Y-axis shows both AP3 values and computed PD values on the test set), we observe that a reduction in KL-divergence of one method compared to another corresponds to a similar pattern in their PDs when meeting the assumptions in Lemma 2. However, some observations deviate from Lemma 2 due to AP3 values exceeding the threshold of $2$.

This divergence is evident in the highest magnitude pruning of the first row of Fig. 4 (left) and $80\%$ lowest magnitude in the second row of Fig. 4 (right).

## 5.4 COMPARING AP2 AND AP3

Our goal is to compare AP2 and AP3 metrics to distinguish lowest, highest, and random pruning methods. Both metrics share the same covariance matrix for a fair comparison. We evaluate identical pruning percentages side by side using AP2 and AP3. Consequently, by leveraging AP2, we can gain

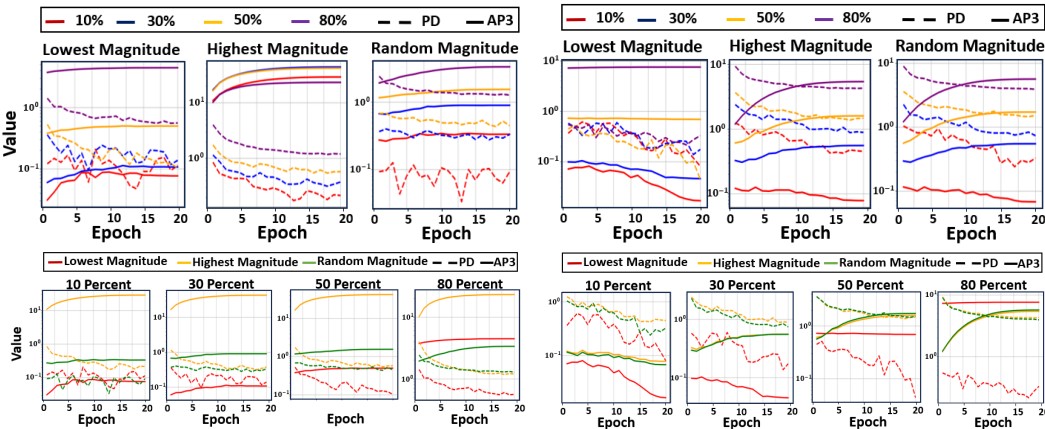

Figure 4: Comparing AP3 & PD for pruned ResNet50. Top- Comparing four pruning percentages for a given pruning method. Bottom-Comparing three pruning methods for a given pruning %. Expriments are done on CIFAR10 (Left) and CIFAR100 (Right).

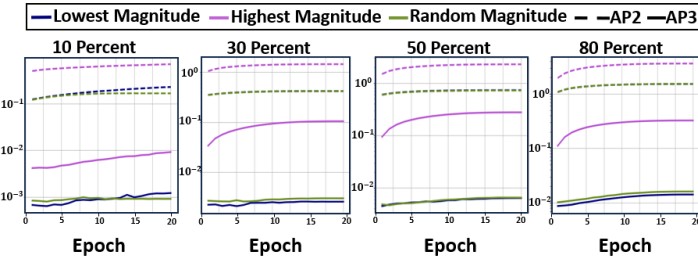

Figure 5: Comparing AP3 and AP2 (Y-axis shows both values) for pruned VGG16 on CIFAR10.

a more comprehensive insight into which pruning method exhibits superior performance differences. The same comparison for pruned AlexNet is shown in Appendix.

It's worth mentioning that there is potential for enhancing AP3 over AP2 either by adjusting covariance matrices within the multivariate T-Student distribution framework or by changing the distribution. Careful adjustments could lead to superior performance for AP3 in specific contexts, opening up an intriguing avenue for further investigation.

## 6 CONCLUSION

In this study, we investigated network sparsity interpretability by employing probabilistic latent spaces. Our main aim was to explore the concept of network pruning using the Kullback-Leibler (KL) divergence of two probabilistic distributions associated with the original network and its pruned counterpart, utilizing their weight tensors as parameters for the latent spaces. Throughout a series of experiments, we demonstrated the efficacy of two projective patterns, namely AP2 and AP3, in elucidating the sparsity of weight matrices. Our analysis provided valuable explanatory insights into the distributions of pruned and unpruned networks. Remarkably, our results indicated that, despite certain limitations encountered in our experiments compared to theoretical claims, nearly all three pruning methods we employed exhibited alignment with the expectations suggested by Lemma 2 when we use the nearly optimal sparsity level.

**Going ahead**, we intend to thoroughly investigate the implications of optimal pruning for preserving strong information flow Ganesh et al. (2022; 2021); Andle & Yasaei Sekeh (2022) between layers through the use of probabilistic latent spaces. Our main goal is to substantiate the claim that optimal pruning ensures the retention of highly comparable mutual information between the layers in the pruned and original networks. By this, we intend to use our findings in this work to better tackle the information transfer challenges in domain adaptation problems.

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

# A  APPENDIX

## A.1  PROOFS

### A.1.1  PROOF OF LEMMA 2

To begin, using loss function in Eq. 1, mentioned in the main paper,

$$\mathbb{E}_{(\mathbf{X},Y)\sim D} \left\{ L(F^{(L)}(\mathbf{X}), Y) \right\} = \mathbb{E}_{(\mathbf{X},Y)\sim D} \left\{ Y \cdot \ell(\omega^*) \right\}, \tag{11}$$

where

$$\ell(\omega^*) := - \sum_{F \in \mathcal{F}} w_F^* \cdot F^{(L)}(\mathbf{X}), \tag{12}$$

let us simplify performance difference $D(F^{(L)}, \widetilde{F}^{(L)})$ defined in Eq. 16 in the main paper.

$$\begin{aligned} D(F^{(L)}, \widetilde{F}^{(L)}) = |\mathbb{E}_{(\mathbf{X},Y)\sim D} \left\{ Y \cdot \ell(\omega^*) \right\} \\ - \mathbb{E}_{(\mathbf{X},Y)\sim D} \left\{ Y \cdot \ell(\widetilde{\omega}^*) \right\} |. \end{aligned} \tag{13}$$

Apply the Jensen inequality, we bound (13) by

$$D(F^{(L)}, \widetilde{F}^{(L)}) \leq \mathbb{E}_{(\mathbf{X},Y)\sim D} \left\{ Y \cdot |\ell(\omega^*) - \ell(\widetilde{\omega}^*)| \right\}. \tag{14}$$

Using quadratic Taylor approximation of $\ell$, we have

$$\begin{aligned}
\|\ell(\omega^*) - \ell(\widetilde{\omega}^*)\|_2 &= \nabla\ell(\widetilde{\omega}^*)^T(\omega^* - \widetilde{\omega}^*) \\
&\quad + \frac{1}{2}(\omega^* - \widetilde{\omega}^*)^T \nabla^2\ell(\widetilde{\omega}^*)(\omega^* - \widetilde{\omega}^*) \\
&\leq \frac{\lambda_{\max}}{2}\|\omega^* - \widetilde{\omega}^*\|_2^2
\end{aligned} \tag{15}$$

where $\lambda_{max}$ is the largest eigenvalue of the Hessian matrix.
Next recall KL definition, defined in Definition 1, which can bounded based on Corollary 1 by

$$\frac{a^T a}{2(tr(\Sigma) + tr(\widetilde{\Sigma})) + a^T a},$$

where $a = \omega^* - \widetilde{\omega}^*$. Hence for $\epsilon < 2$,

$$\|\omega^* - \widetilde{\omega}^*\|_2^2 \leq \frac{\sqrt{\frac{\epsilon}{2}}C}{1 - \sqrt{\frac{\epsilon}{2}}}, \tag{16}$$

where $C = tr(\Sigma) + tr(\widetilde{\Sigma})$.

$$\tag{17}$$

Therefore, based on Equations 15 and 16,

$$D(F^{(L)}, \widetilde{F}^{(L)}) \leq \mathbb{E}_{(\mathbf{X},Y)\sim D} \left\{ Y \cdot \left( \frac{\lambda_{max}\sqrt{\frac{\epsilon}{2}}C}{2(1 - \sqrt{\frac{\epsilon}{2}})} \right) \right\}, \tag{18}$$

where $C = tr(\Sigma) + tr(\widetilde{\Sigma})$.

### A.1.2  PROOF OF COROLLARY 2

Recall (15) from the proof of Lemma 2:

$$\|\ell(\omega^*) - \ell(\widetilde{\omega}^*)\|_2 \leq \frac{\lambda_{\max}}{2}\|\omega^* - \widetilde{\omega}^*\|_2^2 \tag{19}$$

where $\lambda_{max}$ is the largest eigenvalue of the Hessian matrix. Based on Eq. 9 in the main paper, the KL divergence between two Gaussian distribution $G(\omega^{*(L)}, \Sigma)$ and $G(\widetilde{\omega}^{*(L)}, \Sigma)$ is given by

$$\begin{aligned}
&KL(P_{\omega^{*(L)}} \| P_{\widetilde{\omega}^{*(L)}}) \\
&= \frac{1}{2} \left( K + (\omega^{*(L)} - \widetilde{\omega}^{*(L)})^T \Sigma^{-1}(\omega^{*(L)} - \widetilde{\omega}^{*(L)}) \right) \\
&\geq \frac{1}{2} \left( K + \lambda_{min}\|\omega^{*(L)} - \widetilde{\omega}^{*(L)}\|_2^2 \right),
\end{aligned}$$

where $K = 0$ by considering the same covariance matrices for both latent spaces and $\lambda_{min}$ is the minimum eigenvalue of matrix $\Sigma^{-1}$. Using AP3 assumption implies that for small $\epsilon^{(L)}$

$$\frac{1}{2} \left( \lambda_{min}\|\omega^{*(L)} - \widetilde{\omega}^{*(L)}\|_2^2 \right) \leq \epsilon^{(L)},$$

equivalently since $\Sigma^{-1}$ is a positive definite matrix with positive eigenvalues we have

$$\|\omega^{*(L)} - \widetilde{\omega}^{*(L)}\|_2 \leq \sqrt{\frac{2\epsilon^{(L)}}{\lambda_{min}}}. \tag{20}$$

Therefore the RHS of (19) is bounded by

$$\|\ell(\omega^*) - \ell(\widetilde{\omega}^*)\|_2 \leq \frac{\lambda_{max}}{2}\sqrt{\frac{2\epsilon^{(L)}}{\lambda_{min}}}. \tag{21}$$

Going back to (13), this implies

$$D(F^{(L)}, \widetilde{F}^{(L)}) \leq \mathbb{E}_{(\mathbf{X},Y)\sim D}\left\{Y \cdot \left(\frac{\lambda_{max}}{2}\sqrt{\frac{2\epsilon^{(L)}}{\lambda_{min}}}\right)\right\}, \tag{22}$$

Since the covariance matrix $\Sigma$ is assumed to be positive definite, all of its eigenvalues are strictly greater than zero. So, the given bound is an increasing function and this concludes of the proof.

### A.1.3 PROOF OF LEMMA 3

After applying projection map $\mathcal{P}$ the latent variable $Z^{(l+1)} \in \mathcal{Z}$ has a distribution $P_{\omega^{*(l+1)}}$ with parameter vector $\omega^{*(l+1)}$. Similarly, the latent variable $\widetilde{Z}^{(l+1)} \in \mathcal{Z}$ has a distribution $P_{\widetilde{\omega}^{*(l+1)}}$ with parameter vector $\widetilde{\omega}^{*(l+1)}$. To show that

$$KL(P_{\omega^{*(l+1)}}\|P_{\widetilde{\omega}^{*(l+1)}}) \leq \epsilon^{(l+1)}, \tag{23}$$

we equivalently show that

$$KL(P_{\omega^{*(l+1)}}\|P_{\widetilde{\omega}^{*(l+1)}}) = \mathbb{E}_{P_{\omega^{*(l+1)}}}\left[\log\frac{P_{\omega^{*(l+1)}}}{P_{\widetilde{\omega}^{*(l+1)}}}\right] \leq \epsilon^{(l+1)}. \tag{24}$$

Acccording to Definition 4 and Lemma 1,

$$KL(P_{\omega^{*(l)}}\|P_{\widetilde{\omega}^{*(l)}}) \geq \frac{1}{2}\left(\int_{z\in\mathcal{Z}}\left|P_{\omega^{*(l)}}(z) - P_{\widetilde{\omega}^{*(l)}}(z)\right|dz\right)^2. \tag{25}$$

Because of assumption AP3 for layer $l$ and its sparse version for given $\epsilon^{(l)}$ we have

$$KL(P_{\omega^{*(l)}}\|P_{\widetilde{\omega}^{*(l)}}) \leq \epsilon^{(l)}, \tag{26}$$

By using the lower bound in (25), we can write

$$\int_{z\in\mathcal{Z}}\left|P_{\omega^{*(l)}}(z) - P_{\widetilde{\omega}^{*(l)}}(z)\right|dz \leq \sqrt{2\epsilon^{(l)}}. \tag{27}$$

Following the analogous arguments in proof of Lemma 2 and (16) this implies that

$$\|\omega^{*(l)} - \widetilde{\omega}^{*(l)}\|_2^2 \leq \frac{\sqrt{\frac{\epsilon}{2}}C}{1 - \sqrt{\frac{\epsilon}{2}}}, \quad \text{for constant } C \tag{28}$$

On the other hand

$$KL(P_{\omega^{*(l+1)}}\|P_{\widetilde{\omega}^{*(l+1)}}) \leq D_{\chi^2}(P_{\omega^{*(l+1)}}, P_{\widetilde{\omega}^{*(l+1)}})$$
$$= \int_{z\in\mathcal{Z}}\frac{(P_{\omega^{*(l+1)}}(z) - P_{\widetilde{\omega}^{*(l+1)}}(z))^2}{P_{\widetilde{\omega}^{*(l+1)}}(z)}\,dz, \tag{29}$$

where $D_{\chi^2}$ refers to $\chi^2$ divergence. Since $P_{\omega^{*(l)}}$ is a Lipschitz function in $\omega^{(l)}$ for $l = 1,\ldots,L$, therefore there exists a constant $\bar{C}_{l+1}$ such that

$$\left|P_{\omega^{*(l+1)}} - P_{\widetilde{\omega}^{*(l+1)}}\right| \leq \bar{C}_{l+1}\|\omega^{*(l+1)} - \widetilde{\omega}^{*(l+1)}\|_2. \tag{30}$$

Going back to bound in (29), this implies that

$$KL(P_{\omega^{*(l+1)}}\|P_{\widetilde{\omega}^{*(l+1)}}) \leq \int_{z\in\mathcal{Z}}\frac{\bar{C}_{l+1}\|\omega^{*(l+1)} - \widetilde{\omega}^{*(l+1)}\|^2}{P_{\widetilde{\omega}^{*(l+1)}}}\,dz. \tag{31}$$

In addition, we can write

$$\|\omega^{*(l+1)} - \widetilde{\omega}^{*(l+1)}\|^2 \leq \|\omega^{*(l+1)} - \omega^{*(l)}\|^2$$
$$+ \|\omega^{*(l)} - \widetilde{\omega}^{*(l)}\|^2 \tag{32}$$
$$+ \|\widetilde{\omega}^{*(l+1)} - \widetilde{\omega}^{*(l)}\|^2.$$

On the other hand,

$$
\begin{aligned}
\|\omega^{*(l+1)} - \omega^{*(l)}\|^2 &\le \|\omega^{*(l+1)} + \omega^{*(l)}\|^2 \\
&\le \|\omega^{*(l+1)}\|^2 + \|\omega^{*(l)}\|^2 \\
&\quad + 2(\omega^{*(l+1)})^T(\omega^{*(l)}) \\
&\le \|\omega^{*(l+1)}\|^2 + \|\omega^{*(l)}\|^2 \\
&\quad + 2\|\omega^{*(l+1)}\| \, \|\omega^{*(l)}\| \cos(\theta) \\
&\le \|\omega^{*(l+1)}\|^2 + \|\omega^{*(l)}\|^2 \\
&\quad + 2\|\omega^{*(l+1)}\| \, \|\omega^{*(l)}\| \\
&\le c_{\omega^{*(l+1)}} + c_{\omega^{*(l)}} + 2 c_{\omega^{*(l+1)}} c_{\omega^{*(l)}}.
\end{aligned}
\tag{33}
$$

Recall (28) and combining with (33), we bound (32) by

$$
\begin{aligned}
\|\omega^{*(l+1)} - \widetilde{\omega}^{*(l+1)}\|^2 &\le \frac{\sqrt{2\epsilon^{(l)}}}{C_l \mathrm{Vol}(\mathcal{Z})} + c_{\omega^{*(l+1)}} + c_{\omega^{*(l)}} \\
&\quad + 2 c_{\omega^{*(l+1)}} c_{\omega^{*(l)}} + \widetilde{c}_{\omega^{*(l+1)}} + \widetilde{c}_{\omega^{*(l)}} \\
&\quad + 2 \widetilde{c}_{\omega^{*(l+1)}} \widetilde{c}_{\omega^{*(l)}}.
\end{aligned}
\tag{34}
$$

Equivalently

$$
\|\omega^{*(l+1)} - \widetilde{\omega}^{*(l+1)}\|^2 \le \frac{\sqrt{2\epsilon^{(l)}}}{C_l \mathrm{Vol}(\mathcal{Z})} + C_\omega^{(l,l+1)},
\tag{35}
$$

where $C_\omega^{(l,l+1)} = c_{\omega^{*(l+1)}} + c_{\omega^{*(l)}} + 2 c_{\omega^{*(l+1)}} c_{\omega^{*(l)}} + \widetilde{c}_{\omega^{*(l+1)}} + \widetilde{c}_{\omega^{*(l)}} + 2 \widetilde{c}_{\omega^{*(l+1)}} \widetilde{c}_{\omega^{*(l)}}$.

### A.1.4 PROOF OF COROLLARY 3

After applying projection map $\mathcal{P}$ and given that latent variable $Z^{(l+1)} \in \mathcal{Z}$ has Gaussian distribution $P_{\omega^{*(l+1)}} \sim G(\omega^{*(l+1)}, \Sigma)$ and latent variable $\widetilde{Z}^{(l+1)} \in \mathcal{Z}$ has Gaussian distribution $P_{\widetilde{\omega}^{*(l+1)}} \sim G(\widetilde{\omega}^{*(l+1)}, \Sigma)$, we can bound the KL divergence between two Gaussian distribution by

$$
\begin{aligned}
KL(P_{\omega^{*(l+1)}} \| P_{\widetilde{\omega}^{*(l+1)}}) &= \frac{1}{2} \Big( C_1 + (\omega^{*(l+1)} - \widetilde{\omega}^{*(l+1)})^T \\
&\qquad\quad \Sigma^{-1}(\omega^{*(l+1)} - \widetilde{\omega}^{*(l+1)}) \Big) \\
&\le \frac{1}{2} \Big( C_1 + \lambda_{max} \|\omega^{*(l+1)} - \widetilde{\omega}^{*(l+1)}\|_2^2 \Big),
\end{aligned}
\tag{36}
$$

where $\lambda_{max}$ is the maximum eigenvalue of matrix $\Sigma^{-1}$ and $C_1$ is constant. Therefore similar to (33) we have

$$
\begin{aligned}
\|\omega^{*(l+1)} - \widetilde{\omega}^{*(l+1)}\|^2 &\le \|\omega^{*(l+1)} - \omega^{*(l)}\|^2 \\
&\quad + \|\omega^{*(l)} - \widetilde{\omega}^{*(l)}\|^2 \\
&\quad + \|\widetilde{\omega}^{*(l+1)} - \widetilde{\omega}^{*(l)}\|^2.
\end{aligned}
\tag{37}
$$

Additionally, from the assumption AP3 for layer $l$ that is

$$
KL(P_{\omega^{*(l)}} \| P_{\widetilde{\omega}^{*(l)}}) \le \epsilon^{(l)},
\tag{38}
$$

and (20) in the proof of Corollary 2 we have

$$
\|\omega^{*(l)} - \widetilde{\omega}^{*(l)}\|_2 \le \sqrt{\frac{2\epsilon^{(l)} - K}{\lambda_{min}}}, \qquad \text{for constant } K,
\tag{39}
$$

where $\lambda_{min}$ is the minimum eigenvalue of matrix $\Sigma^{-1}$. Combining (33) and (39), we get

$$
\begin{aligned}
\|\omega^{*(l+1)} - \widetilde{\omega}^{*(l+1)}\|^2 &\le \sqrt{\frac{2\epsilon^{(l)} - K}{\lambda_{min}}} + c_{\omega^{*(l+1)}} + c_{\omega^{*(l)}} \\
&\quad + 2 c_{\omega^{*(l+1)}} c_{\omega^{*(l)}} + \widetilde{c}_{\omega^{*(l+1)}} + \widetilde{c}_{\omega^{*(l)}} \\
&\quad + 2 \widetilde{c}_{\omega^{*(l+1)}} \widetilde{c}_{\omega^{*(l)}}.
\end{aligned}
\tag{40}
$$

Going back to (36) and (40), we have

$$KL(P_{\omega^{*(l+1)}} \| P_{\widetilde{\omega}^{*(l+1)}}) \leq \frac{1}{2} \Big( C_1 + \lambda_{max} C_\omega^{(l,l+1)} \Big)$$
$$\big( \frac{2\epsilon^{(l)} - K}{\lambda_{min}} \big), \tag{41}$$

where $C_\omega^{(l,l+1)} = c_{\omega^{*(l+1)}} + c_{\omega^{*(l)}} + 2c_{\omega^{*(l+1)}} c_{\omega^{*(l)}} + \widetilde{c}_{\omega^{*(l+1)}} + \widetilde{c}_{\omega^{*(l)}} + 2\widetilde{c}_{\omega^{*(l+1)}} \widetilde{c}_{\omega^{*(l)}}$ and $K$ is a constants. This concludes corollary 3 when $\epsilon^{(l+1)} := \frac{1}{2} \Big( C_1 + \lambda_{max} C_{\sigma^{-1},\sigma,x}^2 \big( \frac{2\epsilon^{(l)} - K}{\lambda_{min}} \big) \Big)$ and proof is completed.

### A.1.5 PROOF OF THEOREM 2

Going back to (25) and considering Corollary 1, we have:

$$KL(P_{\omega^*} \| P_{\widetilde{\omega}^*}) \geq 2 \big( \frac{\|\omega^* - \widetilde{\omega}^*\|^2}{2(tr(\Sigma_1) + tr(\Sigma_2)) + \|\omega^* - \widetilde{\omega}^*\|^2} \big)^2. \tag{42}$$

Further, according to (29) and (30), we have:

$$-KL(P_{\widetilde{\omega}^*} \| P_{\widetilde{\omega}^t}) \geq - \int_{z \in \mathcal{Z}} \frac{(P_{\widetilde{\omega}^*}(z) - P_{\widetilde{\omega}^t}(z))^2}{P_{\widetilde{\omega}^t}(z)} \, dz$$
$$\geq -(C^t)^2 \int_{z \in \mathcal{Z}} \frac{\|\widetilde{\omega}^* - \widetilde{\omega}^t\|_2^2}{P_{\widetilde{\omega}^t}(z)} dz \tag{43}$$
$$\geq -\overline{C}^t \|\widetilde{\omega}^* - \widetilde{\omega}^t\|_2^2,$$

the last inequality on the RHS above is derived because of the assumption that $\mathbb{E}[1/(P_{\widetilde{\omega}^t})^2]$ is bounded say by constant $C$. Here $\overline{C}^t = (C^t)^2 C$. Combine (42) and (43), we have

$$KL(P_{\omega^*} \| P_{\widetilde{\omega}^*}) - KL(P_{\widetilde{\omega}^*} \| P_{\widetilde{\omega}^t}) \geq$$
$$2 \big( \frac{\|\omega^* - \widetilde{\omega}^*\|^2}{2(tr(\Sigma_1) + tr(\Sigma_2)) + \|\omega^* - \widetilde{\omega}^*\|^2} \big)^2 - \overline{C}^t \|\widetilde{\omega}^* - \widetilde{\omega}^t\|_2^2. \tag{44}$$

Under the assumption that $F^{(L)}$ and $\widetilde{F}_t^{(L)}$ have approximately AP3 in iteration $t$ given in Eq. 26 in the main paper we bound the RHS of (44):

$$\Big| \overline{K}_1 \|\omega^* - \widetilde{\omega}^*\|_2^2 - \overline{K}_2^t \|\omega^* - \widetilde{\omega}^t\|_2^2 \Big| \leq \epsilon_t. \tag{45}$$

Denote function $f$ as

$$f(\omega) := \mathbb{E}_{(\mathbf{X},Y) \sim D} \left[ Y.\ell(\omega) \right].$$

We aim to show that

$$|f(\omega^*) - f(\widetilde{\omega}^t)| = O(\frac{\epsilon_t}{t}).$$

Using quadratic Taylor approximation, we have

$$\|f(\widetilde{\omega}^t) - f(\omega^*)\|_2 = \nabla f(\omega^*)^T (\widetilde{\omega}^t - \omega^*)$$
$$+ \frac{1}{2} \nabla^2 f(\omega^*) \|\widetilde{\omega}^t - \omega^*\|_2^2$$
$$\leq \frac{\lambda_{\max}}{2} \|\widetilde{\omega}^t - \omega^*\|_2^2 \tag{46}$$
$$\leq \frac{\lambda_{\max}}{2} \|\widetilde{\omega}^t - \widetilde{\omega}^* + \widetilde{\omega}^* - \omega^*\|_2^2$$
$$\leq \frac{\lambda_{\max}}{2} \|\widetilde{\omega}^t - \widetilde{\omega}^*\|_2^2 + \frac{\lambda_{\max}}{2} \|\widetilde{\omega}^* - \omega^*\|_2^2$$

it is enough to find an upper bound for $\|\widetilde{\omega}^t - \widetilde{\omega}^*\|_2^2$ and $\|\widetilde{\omega}^* - \omega^*\|_2^2$. Regarding $\|\widetilde{\omega}^t - \widetilde{\omega}^*\|_2^2$, based on gradient updates, we have

$$\widetilde{\omega}^* = \widetilde{\omega}^t - \alpha_t(\nabla f(\widetilde{\omega}^t)) - \ldots - \alpha_{T-1}(\nabla f(\widetilde{\omega}^{T-1})) \tag{47}$$

Without loss of generality suppose that for any $i, j \in \{t, \ldots, T-1\}$, $\alpha_i = \alpha_j = \alpha$. Therefore,

$$
\begin{aligned}
\|\widetilde{\omega}^* - \widetilde{\omega}^t\|_2^2 &= \alpha^2 \| \sum_{i=t}^{T-1} \nabla f(\widetilde{\omega}^i) \|_2^2 \\
&\leq \alpha_2 \sum_{i=t}^{T-1} \|\nabla f(\widetilde{\omega}^i)\|_2^2 \\
&\leq \alpha^2 (T-t) \max_{i=t\ldots,T-1} (\|\nabla f(\widetilde{\omega}^i)\|_2^2) \\
&\leq (T-t) C_{\alpha, \max_{t,\ldots,T-1}}(\nabla)
\end{aligned}
\tag{48}
$$

Going back to (45),

$$
\left| \overline{K}_1 \|\omega^* - \widetilde{\omega}^*\|_2^2 - \overline{K}_2^t \|\widetilde{\omega}^* - \widetilde{\omega}^t\|^2 \right| \leq \epsilon_t
\tag{49}
$$

Hence,

$$
-\epsilon_t \leq \overline{K}_1 \|\omega^* - \widetilde{\omega}^*\|_2^2 - \overline{K}_2^t \|\widetilde{\omega}^* - \widetilde{\omega}^t\|^2 \leq \epsilon_t
\tag{50}
$$

Therefore,

$$
\begin{aligned}
\overline{K}_1 \|\omega^* - \widetilde{\omega}^*\|_2^2 - \overline{K}_2^t (T-t) C_{\alpha, \max_{t,\ldots,T-1}}(\nabla) \\
\leq \overline{K}_1 \|\omega^* - \widetilde{\omega}^*\|_2^2 - \overline{K}_2^t \|\widetilde{\omega}^* - \widetilde{\omega}^t\|^2 \leq \epsilon_t
\end{aligned}
\tag{51}
$$

Finally,

$$
\|\omega^* - \widetilde{\omega}^*\|_2^2 \leq \frac{\overline{K}_2^t (T-t) C_{\alpha, \max_{t,\ldots,T-1}}(\nabla) + \epsilon_t}{\overline{K}_1}
\tag{52}
$$

Going back to 46,

$$
\begin{aligned}
\|f(\widetilde{\omega}^t) - f(\omega^*)\|_2 &\leq \frac{\lambda_{\max}}{2} (T-t) C_{\alpha, \max_{t,\ldots,T-1}}(\nabla) \\
&\quad + \frac{\lambda_{\max}}{2} \|\widetilde{\omega}^* - \omega^*\|_2^2 \\
&\leq \frac{\lambda_{\max}}{2} (T-t) C_{\alpha, \max_{t,\ldots,T-1}}(\nabla) \\
&\quad + \frac{\lambda_{\max}}{2} \left( \frac{\overline{K}_2^t (T-t) C_{\alpha, \max_{t,\ldots,T-1}}(\nabla) + \epsilon_t}{\overline{K}_1} \right) \\
&\leq C_1 (T-t) + C_2 \epsilon_t = g(\epsilon_t, T)
\end{aligned}
\tag{53}
$$

Note that for smaller $\epsilon_t$ and larger $t$, the bound in 34 becomes tighter.

## A.2 ADDITIONAL EXPERIMENTS

Our previous results in the main paper has highlighted the reliability of Lemma 2. This underscores the fact that as the AP2/AP3 values of one pruning method decrease in relation to another, a coherent and predictable pattern surfaces in the behavior of their corresponding performance differences. As a supplement, we provide additional results obtained by using three benchmark pre-trained models: AlexNet, ResNet50, and VGG16 on two common datasets CIFAR10 and CIFAR100 to have a comparison of AP2 and AP3 metrics with performance differences.

This section unfolds into three distinct segments. In section A.2.1, we embark on a comprehensive comparison of AP2 (AP3) alongside the computed performance difference on the training set. This analysis encompasses each model's behavior on the CIFAR10 and CIFAR100 datasets. Section A.2.2 delves into exploring the relationship between AP3 and the performance differences (PDs), computed on both training and test sets, between an original network and its sparse counterpart. This investigation is focused on the final layer of the network and employs a multivariate Gaussian distribution with a non-diagonal covariance matrix. Section A.2.3 introduces a continuous examination of the comparison between AP3 and PD employing the multivariate T-student, which was conducted on ResNet50 in the main paper. Our focus in section A.2.3, is directed towards two models: AlexNet and VGG16. In section A.2.4, two metrics AP2 and AP3 are compared together.

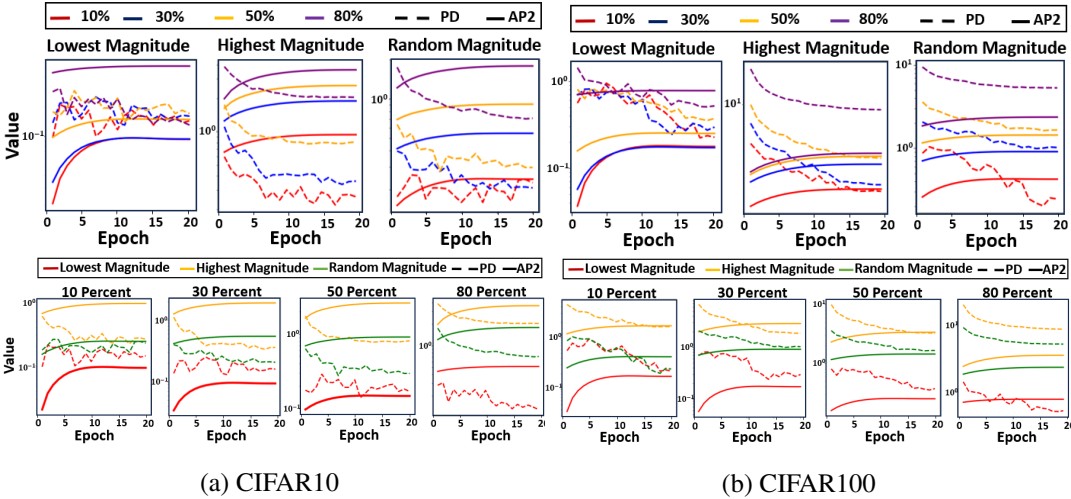

Figure 6: Top-Comparing AP2 & computed PD on the test set for pruned AlexNet. Bottom-Comparing three pruning methods for a given pruning %.

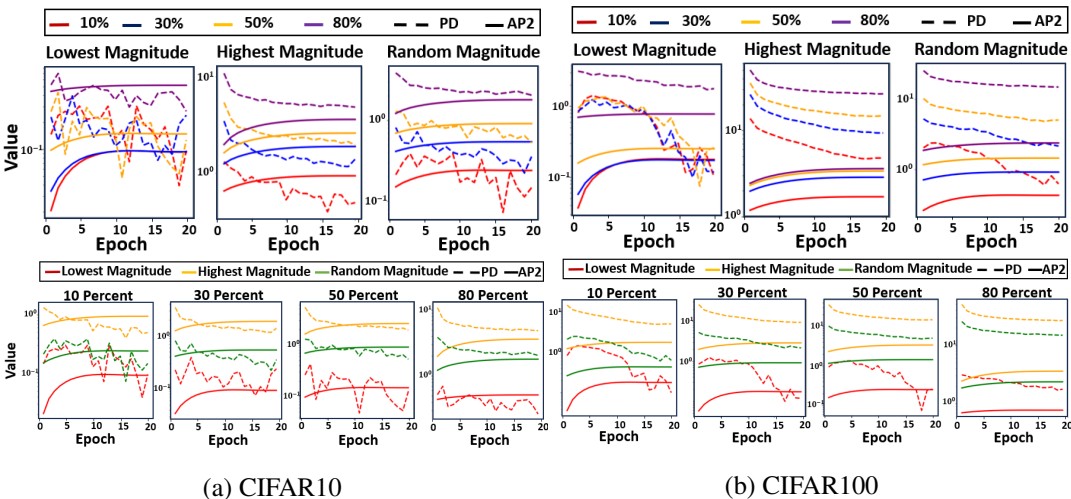

Figure 7: Top-Comparing AP2 & computed PD on the training set for pruned AlexNet. Bottom-Comparing three pruning methods for a given pruning %.

### A.2.1 MULTIVARIATE GAUSSIAN DISTRIBUTION WITH DIAGONAL COVARIANCE MATRICES

In this section, our objective is to present a more comprehensive set of experiments that delve into the comparison between AP2 and PDs on both training set and test set. Note that in Fig 6, Y-axis shows the value of both AP2 and PD, computed on the test set while in Figs. 7-10 Y-axis shows the value of both AP2 and computed PD on training set.

**AlexNet** A comparison of AP2 metric and PD on either test set or training set using the AlexNet model, applied to CIFAR10 and CIFAR100 datasets, and its sparse version are shown in Figs. 6 and 7. These assessments utilize the test set and the training set to compute performance differences, respectilvely. Although Figs. 6 and 7 effectively show the validity of Lemma 2, the lowest magnitude pruning approach encounters occasional hurdles that could be attributed to either notable discrepancies between theoretical assumptions and empirical findings or pruning of predominantly non-essential weights.

**ResNet50** A comparison of various pruning percentages is illustrated in Fig. 8, with the first row depicting the pruning methods and the second-row illustrating different sparsity levels. Similarly

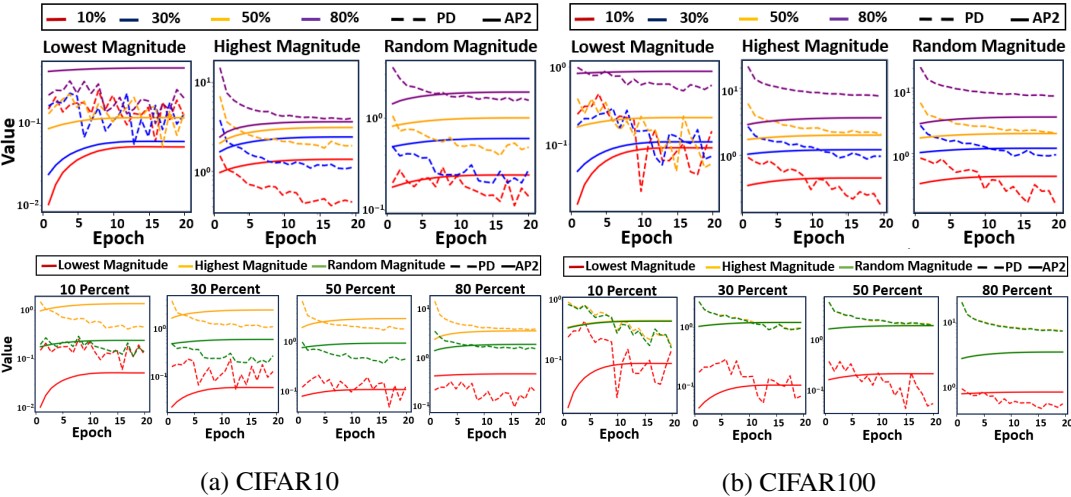

(a) CIFAR10      (b) CIFAR100

Figure 8: Top-Comparing AP2 & computed PD on the training set for pruned ResNet50. Bottom-Comparing three pruning methods for a given pruning %.

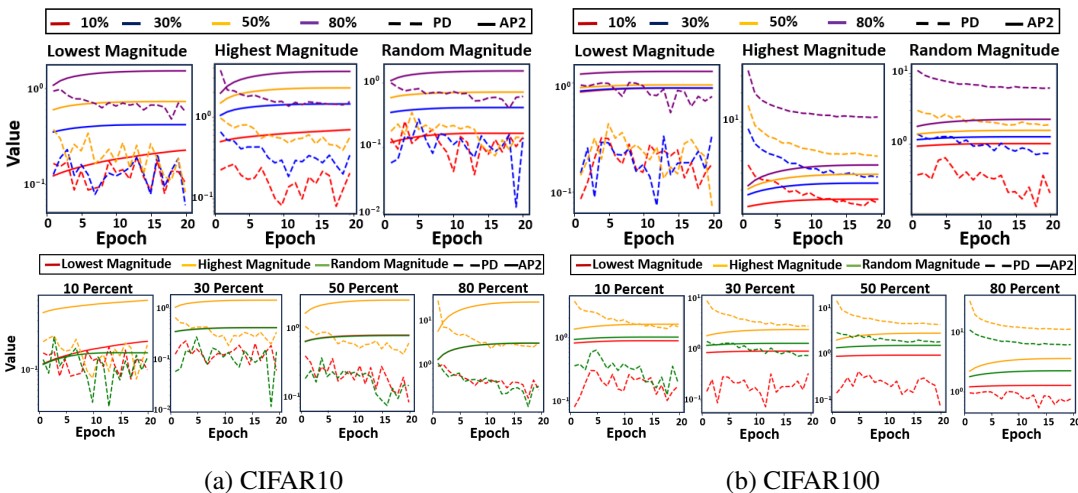

(a) CIFAR10      (b) CIFAR100

Figure 9: Top-Comparing AP2 & computed PD on the training set for pruned VGG16. Bottom-Comparing three pruning methods for a given pruning %.

to the drawbacks mentioned for AlexNet, the results for the lowest magnitude pruning method are not strong evidence of the findings in Lemma 2. However, it's noteworthy that the outcomes of the other two pruning methods are consistent with our expectations, aligning well with the implications of Lemma 2.

**VGG16** Fig. 10 is provided as an investigation of the relationship between AP2 and computed performance differences on the training set using the VGG16 network.

### A.2.2   MULTIVARIATE GAUSSIAN DISTRIBUTION WITH NON-DIAGONAL COVARIANCE MATRICES

As previously demonstrated in section 5.2, we have validated the correctness of Lemma 2 for AP3 using the multivariate T-Student distribution. In this section, we aim to further solidify the reliability of Lemma 2 by conducting experiments with AP3 using multivariate Gaussian distributions with non-diagonal covariance matrices. These experiments are conducted on pruned ResNet50 using the CIFAR10 dataset.

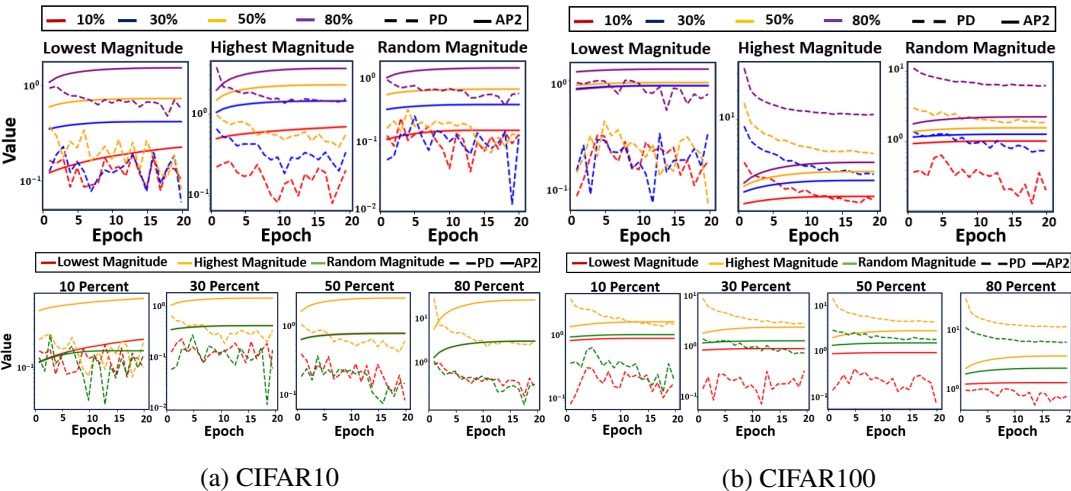

Figure 10: Top-Comparing AP2 & computed PD on the training set for pruned VGG16. Bottom-Comparing three pruning methods for a given pruning %.

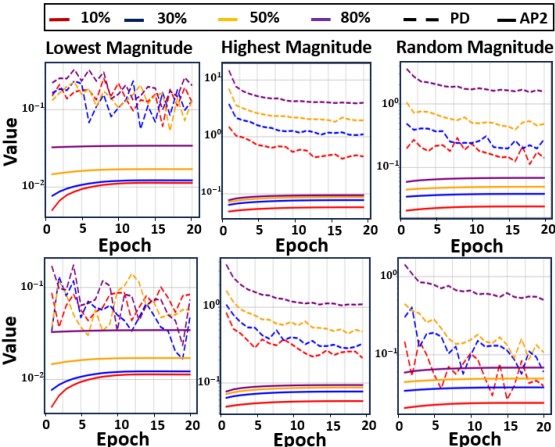

Figure 11: Comparing AP3 and PD for pruned ResNet50 on CIFAR10. Top-PD on the training set. Botton-PD on the test set.

**ResNet50:** Figure 11 visually reinforces the validity of Lemma 2 by employing AP3, using multivariate Gaussian distributions with non-diagonal covariance matrices, on the pruned ResNet50 model with the CIFAR10 dataset. In Figure 11, it's notable that the highest and random magnitude pruning methods consistently align with Lemma 2, verifying its validity. However, the lowest magnitude pruning occasionally presents challenges due to either significant disparities between theoretical assumptions and practical observations, such as using optimal weights in Lemma 2 or removing non-essential weights or variability in stochastic gradient descent. Despite all such challenges, we observed the anticipated behavior described in Lemma 2. This signifies that as one pruning method's KL-divergence decreases relative to another, its performance difference likewise diminishes. Importantly, all methods and percentages shared the same covariance matrices for equitable comparison.

### A.2.3 MULTIVARIATE T-STUDENT DISTRIBUTION ON ALEXNET AND VGG16

Here, we used the multivariate T-Student distribution, with a sample size of 600,000 data points, to compare AP3, derived by utilizing Monte Carlo estimation, and performance differences. We employed the Pyro package, which is backed by PyTorch, to utilize the multivariate T-Student distribution, The choice of using Monte Carlo estimation was primarily due to the limitations of KL packages available in Torch, which only supported specific distributions, mostly univariate distri-

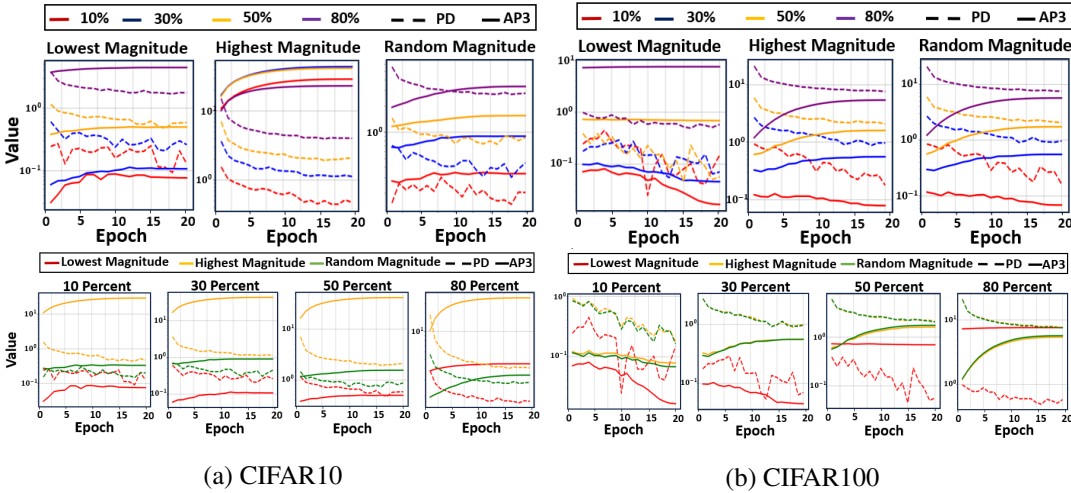

(a) CIFAR10
(b) CIFAR100

Figure 12: Comparing AP3 & PD for pruned ResNet50. Y-axis shows the value of both AP3 and PD on training set.

butions. Results for different benchmarks are as follows. Note that, in Figs. 13-16, the first row shows computed performance differences on the training set, while in the second row, we observe the corresponding results on the test set. The main paper provides an analysis of the results for ResNet50, focusing on the performance differences observed on the test set. Continuing our analysis, in Fig. 12, we extend our investigation to ResNet50 by conducting experiments that consider performance differences on the training set. The results for AlexNet are shown in Figs. 13 and 14,

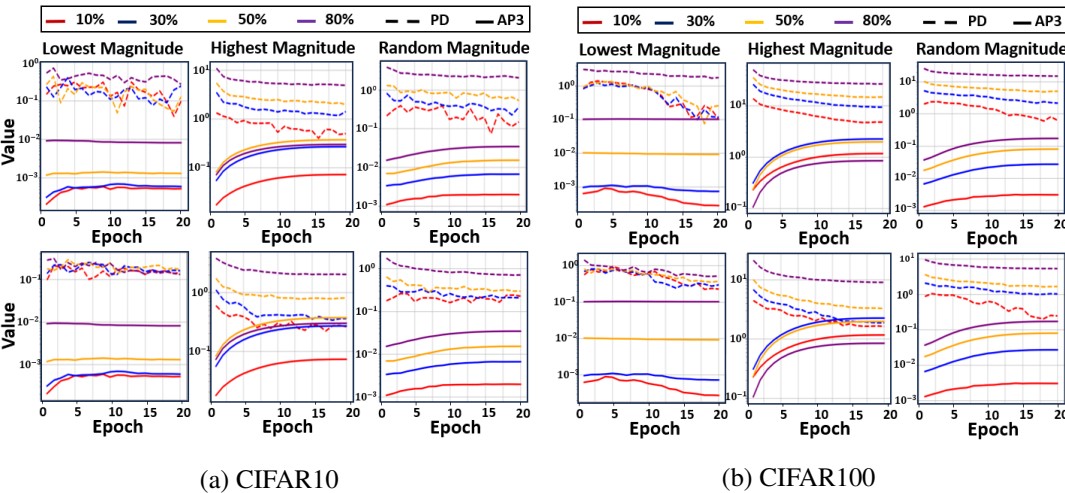

(a) CIFAR10
(b) CIFAR100

Figure 13: Comparing AP3 & PD of different percentages of pruned AlexNet using different pruning methods.

which provide clear evidence supporting Lemma 2, with the exception of the highest magnitude using $80\%$ percentage. One contributing factor to this discrepancy is the size of $m^*$, representing the count of zeros within the mask matrix. Relying on $80\%$ as the optimal pruning percentage might not necessarily be the most effective approach. Note that by adhering to the theorems, we can potentially elucidate the performance difference via KL-divergence, especially if we consider the optimal sparsity levels.

In case of VGG16, our observations are presented in Figs. 15 and 16. Within these figures, specific scenarios come into focus, particularly when applying $80\%$ highest magnitude pruning or random

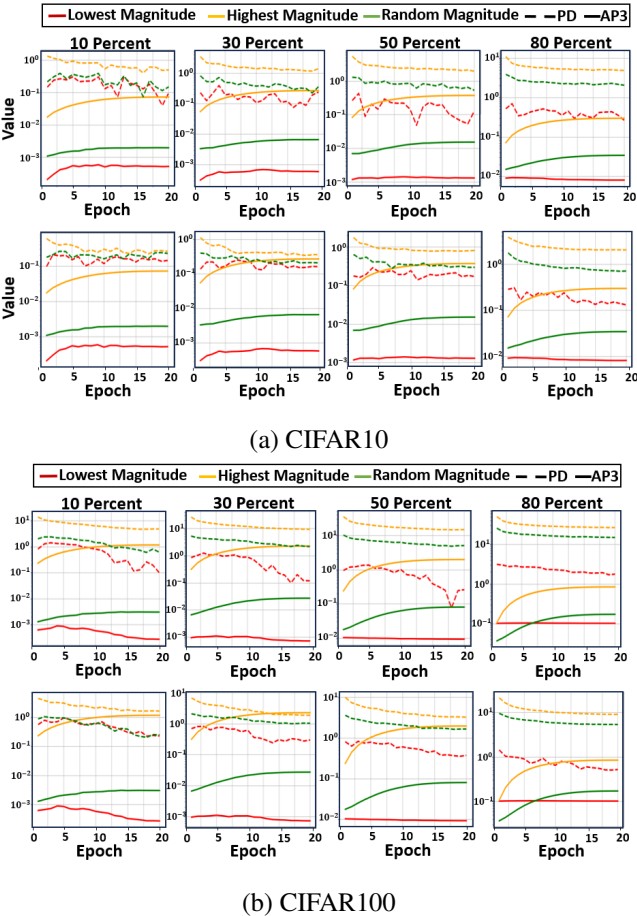

Figure 14: Comparing AP3 & PD for pruned AlexNet using three pruning methods for a given pruning %.

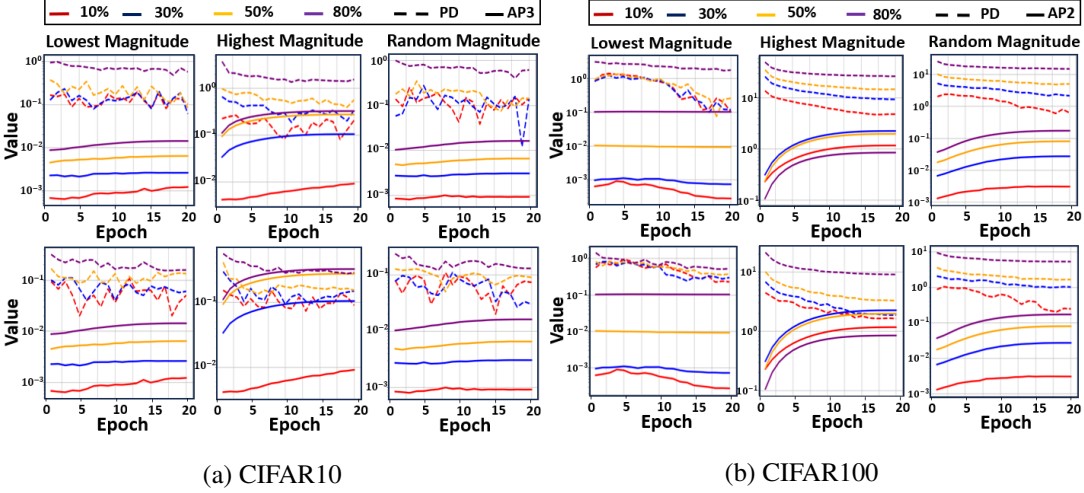

Figure 15: Comparing AP3 & PD of different percentages of pruned VGG16 using different pruning methods.

magnitude pruning. These cases manifest deviations from our anticipated results in Lemma 2, stemming from instances where the mask size exceeds the optimal sparsity threshold.

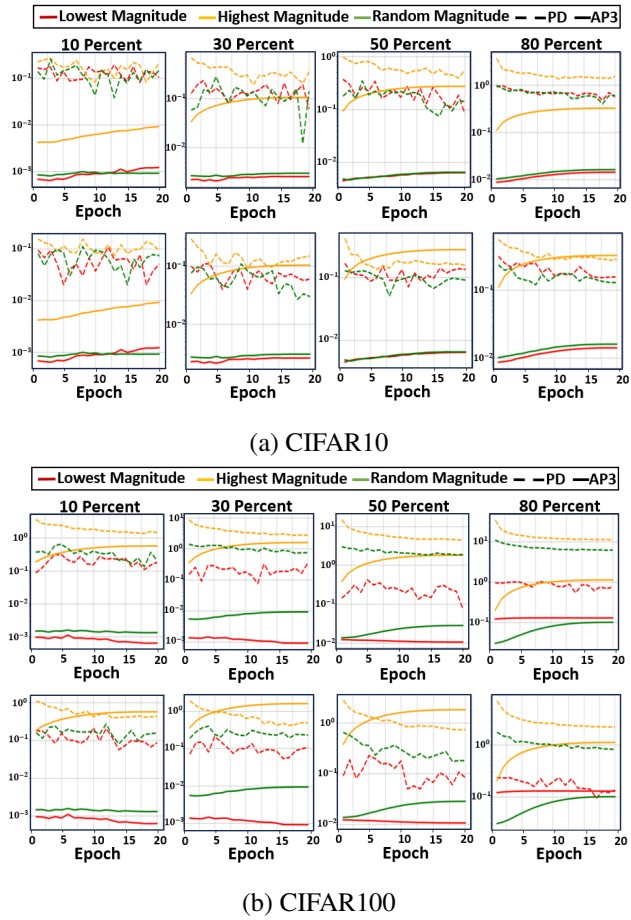

Figure 16: Comparing AP3 & PD of pruned VGG16 using three pruning methods for a given pruning %.

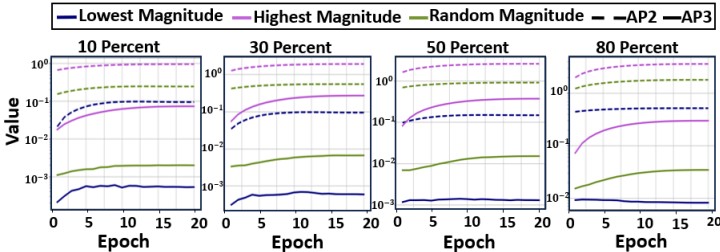

Figure 17: Comparing AP3 and AP2 (Y-axis shows both values) for pruned AlexNet on CIFAR10.

### A.2.4 COMPARING AP2 AND AP3

Our objective is to provide more expriments to assess whether AP2 or AP3 provides better differentiation among various pruning methods, including lowest, highest, and random pruning. We aim to determine which of these patterns, AP2 or Ap3, offers greater insights into which pruning method exhibits superior performance differences. As outlined in Lemma 2, under specific conditions, a lower KL divergence results in lower performance differences. Therefore, a greater contrast between AP2 values (or AP3 values) for two pruning methods gives us greater confidence in discerning their performance differences. Fig 17 shows the compasrison of AP2 and AP3 for pruned Alexnet using CIFAR10.

### A.3 TECHNICAL CONFIGURATION

### A.3.1 COMPUTING RESOURCES

The experiments are conducted using pytorch on a dedicated computing platform that encompassed both CPU and GPU resources. The operating system employed is Microsoft Windows [version 10.0.19045.2846], running on a machine equipped with 16 GB of memory. The central processing unit (CPU) utilized is an Intel64 Family Model 85 Stepping 7. In addition, the graphics processing unit (GPU) utilized was an NVIDIA GeForce RTX 3090, leveraging its high-performance architecture for accelerated computations and complex data processing tasks. This combined computing platform facilitated the execution of the experiments presented in this study. The experiments are conducted using a variety of software packages, each contributing to the analysis and results. The following packages were integral to the research:

- Python: 3.8.10
- Matplotlib: 3.7.1
- Torch: 2.0.0+cu117
- Pandas: 1.5.3
- NumPy: 1.23.5
- Torchvision: 0.15.1+cu117

