# OpenReview forum: "Towards Explaining Deep Neural Network Compression Through a Probabilistic Latent Space"
_ICLR.cc/2024/Conference — Submitted to ICLR 2024_

### Official Review · Reviewer_kKNE · 2023-10-31

**Soundness:** 2 fair
**Presentation:** 1 poor
**Contribution:** 3 good
**Rating:** 5
**Confidence:** 4

**Summary:**

In this paper, the authors have tackled the theoretical understanding of pruning DDNs as a canonical compression technique. In particular, the weights of the originally trained network and the corresponding pruned one are projected to a probabilistic latent space with the same dimension and KL divergence between the distributions of the projections is upper-bounded. In addition, some experiments have been conducted to support the theoretical claims.

**Strengths:**

While almost all compression papers are super practical and do not provide any theoretical justification for why compressed networks have the same or even better generalization,  this paper tackles to theoretically establish an analytical framework for understanding the pruning of DNNs through the projection of weights into a probabilistic latent space. In my opinion, this is the strength of this paper.

**Weaknesses:**

1 - The English of this paper needs to be improved significantly.

2 - Many statements and definitions are unnecessarily wordy and long, making reading and following the paper difficult.

3 - What is the meaning of $\mathcal{F}$ in the emprical risk minimization? I don't follow the notation used in equation (1). if $w_F$ denotes the whole set of weights, what is the meaning of the product between $Y$ and this term (loss is supposed to be a scalar value)? What kind of product is considered in the expression of $l(w)$? The same thing in Lemma 2 for the function $g$,

4 - In the example on page 3, the authors have said that "without loss of generality, assume hat ...". Why should the covariance of the projected pruned weights be positive definite (all eigenvalues are strictly positive ?)? This is actually used in the experiment of section 5.2.

5 - I understand this is a theoretical paper, but choosing only CIFAR10/100 datasets, experimenting in a controlled setup like a diagonal covariance matrix, and on the last layer, using only three architectures for a simple classification problem (no other tasks have been considered), and no experiments for other architectures (RNNs, Transformers, etc) is not completely convincing approach.

6 - I am confused with the notion of iteration $t$ in definition 5, and the "trained sparse network" in the theorem 2, Lemma 2, etc.:
     Typically, pruning is an iterative approach, meaning that when a trained network is pruned, it should be retrained again (or fine-tuned) to
     get back its original performance, and the process may be repeated if further pruning is required. Is this the case for you? if so, what are the
     conditions for the retraining/fine-tuning procedure? (i.e., what is the initialization for the pruned weights? What about many other hyper-
     parameters values? From the experiment, it seems the authors have indeed fine-tuned the pruned network, but there is nothing about
     it in the theoretical claims. In addition, does the cycle of prune-retrain/fine-tune happen only one time?

7- The experiment section is quite confusing. In particular, the explanation of the plots, and legends are not consistent. There are many plots that make them hard to follow. I highly recommend showing 2 or 3 figures with super clear legends and explaining them neatly.

**Questions:**

Please see my comments in the "weakness" part.

---

> ### Author Response · Authors · 2023-11-22
> **Response for Reviewer kKNE**
>
> Thank you for your insightful concerns and feedback on how we could improve our paper. We have taken it all into consideration, and implemented changes where possible and appropriate.
>
> **1. The English of this paper needs to be improved significantly.**
>
> Thank you for your point,  throughout the paper, we have improved the language and provided further clarifications.
>
> **2. Many statements and definitions are unnecessarily wordy and long, making reading and following the paper difficult.**
>
> Thank you for your feedback. We have made revisions throughout the paper.
>
> **3. What is the meaning of $\mathcal{F}$
>  in the empirical risk minimization?**
>
> Thank you for your question.
> $\mathcal{F}$ represents a set of networks, each characterized by a set of parameters denoted as
>  $w_{F}$. The objective of minimizing the loss function is equivalent to finding a network, or more precisely, a set of parameters, that minimizes the value of the formula given specific inputs and labels. In addition, one way to think of loss function in deep neural networks is a correlation between the class label and the last layer of the network. More specifically, training a classifier is defined by minimizing a loss function that decreases with the correlation between the weighted combination of the layers and the class label.
>  In our paper, for simplicity, this correlation is defined as an inner product between $Y$ and the last layer features, but it is straightforward to generalize this to more practical loss functions such as logistic or hinge loss. Note that the same definition is used in the following papers to name a few.
>
>  Ilyas A, Santurkar S, Tsipras D, Engstrom L, Tran B, Madry A. Adversarial examples are not bugs, they are features. Advances in neural information processing systems, 2019.
>
> Craig J, Andle J, Nowak TS, Sekeh SY. A Theoretical Perspective on Subnetwork Contributions to Adversarial Robustness. arXiv preprint arXiv:2307.03803, 2023.
>
> **4. In the example on page 3, the authors have said that "without loss of generality, assume hat ...".**
>
> We thank the reviewer for the constructive feedback. In the revised version, we have made the sentence more explicit to clarify. In this paper, we exclusively utilize positive definite covariance matrices for the latent spaces, and our proofs are based on this assumption. This assumption is now explicitly stated in the introduction for transparency. Additionally, in the example provided, we assume, without loss of generality, that the two covariance matrices are identical. Given the assumption maintained throughout the paper, we can confirm those covariance matrices are positive definit.
>
> **5. I understand this is a theoretical paper, but choosing only CIFAR10/100 datasets, experimenting in a controlled setup like a diagonal covariance matrix, and on the last layer, using only three architectures for a simple classification problem (no other tasks have been considered), and no experiments for other architectures (RNNs, Transformers, etc) is not completely convincing approach.**
>
> Regarding the setup and the use of non-diagonal covariance matrices, we conducted experiments as detailed in the Appendix, specifically in the "MULTIVARIATE GAUSSIAN DISTRIBUTION WITH NON-DIAGONAL COVARIANCE MATRICES" section. While our primary goal was to establish theoretical explanations through our theorems, we acknowledge the limited inclusion of experiments with networks beyond the three utilized in this study.
>
> **6.I am confused with the notion of iteration $t$
>  in definition 5, and the "trained sparse network" in the theorem 2**
>
> In general, the pruning discussed in this paper is characterized as one-shot pruning rather than iterative pruning. This entails initially training the original network to achieve optimality, followed by a one-shot pruning. Subsequently, the pruned network undergoes fine-tuning for a specified number of epochs. In Definition 5, the iteration
> $t$ corresponds to the epoch $t$ during fine-tuning of the pruned network.
>
> Additionally, when referring to a "trained sparse network," we mean the pruned network that is already fine-tuned.
>
> Regarding your question about  "initialization for the pruned weights", if you mean how the network is initilized for the pruning phasse, the relevant details can be found in Section 5.1 (Pruning Phase).  Indeed, the network is initialized with the best parameters derived from training the original model for a specified number of epochs (in this case, 100 epochs).
>
> **7. The experiment section is quite confusing. In particular, the explanation of the plots, and legends are not consistent.**
>
> To balance between extensive experimental validation of our theoretical analysis and clear explanations of results, we included most of our experiments in the Appendix and avoided repeated explanations in the main paper. However, in the revision, we clarified the plots further.

---

### Official Review · Reviewer_EYVP · 2023-11-01

**Soundness:** 3 good
**Presentation:** 3 good
**Contribution:** 3 good
**Rating:** 6
**Confidence:** 3

**Summary:**

The paper delves into the domain of deep neural networks with a specific focus on network pruning and its relationship with probabilistic distributions. The primary objective is to understand and explain the behavior of pruned networks both theoretically and empirically.

The paper introduces intricate mathematical derivations to explain the behavior of pruned networks. A notable highlight is the introduction and exploration of properties like AP2 and AP3, which are closely linked to the Kullback-Leibler (KL) divergence between certain distributions.

The authors establish a connection between network pruning techniques and probabilistic distributions. By utilizing the AP3 property, they explain the training convergence of a compressed network to an optimal sparsity level without compromising performance.

The research showcases a comparative analysis between AP2 and AP3 properties to discern which provides better differentiation among various pruning methods. The aim is to offer insights into the performance differences of these methods.

Empirical evidence is presented through experiments on popular architectures like ResNet50, AlexNet, and VGG16 across datasets such as CIFAR10 and CIFAR100. The results validate the theoretical predictions and provide practical insights into the behavior of pruned networks.

**Strengths:**

The paper introduces novel concepts like the AP2 and AP3 properties, which aim to explain the behavior of pruned deep neural networks through a probabilistic lens.The connection between network pruning and probabilistic distributions, especially in the context of non-Bayesian networks, appears to be a fresh perspective in the domain.

The research provides a balanced mix of theoretical derivations and empirical validations. Mathematical foundations are rigorously laid out, and their implications are tested with real-world experiments. Experimental results on architectures like ResNet50, AlexNet, and VGG16 across datasets such as CIFAR10 and CIFAR100 lend credibility to the theoretical claims.

The paper is structured coherently with distinct sections for mathematical derivations, experimental results, related work, and technical configurations. The document effectively uses figures and graphical representations to aid understanding, especially in the experimental results section.

Network pruning is a crucial area in deep learning, especially given the computational challenges of deploying large models. By providing insights into the behavior of pruned networks and their connection to probabilistic distributions, the paper addresses a significant aspect of model compression and optimization. The findings can potentially guide researchers and practitioners in better understanding and implementing pruning techniques for deep neural networks.

**Weaknesses:**

The focus on specific architectures (ResNet50, AlexNet, VGG16) may limit the general applicability of the findings. The behavior of the pruning methods on newer or different architectures remains unexplored.

The paper could benefit from a direct comparison with state-of-the-art pruning techniques in terms of performance, computational efficiency, and model robustness. This would give readers a clear benchmark on where the introduced methods stand relative to existing methods.

There is no discussion on the sensitivity of the proposed methods to hyperparameters. Understanding how sensitive the AP2 and AP3 properties are to changes in hyperparameters would be vital for practitioners looking to apply these methods.

There is no mention of the robustness of the pruning methods against adversarial attacks or their performance under distribution shifts. Given the increasing importance of model robustness, an assessment in these areas could significantly strengthen the paper.

**Questions:**

Could you provide direct empirical comparisons of the AP2 and AP3 properties against traditional pruning baselines? How do these new methods fare in terms of conventional metrics like accuracy, FLOPs reduction, and memory footprint?

Have you tested the robustness of the pruned networks using AP2 and AP3 against adversarial attacks? Additionally, how do these pruned networks perform under distribution shifts or in out-of-distribution scenarios?

You mentioned deviations from expected results based on Lemma 2. Could you delve deeper into the causes of these discrepancies and potentially adjust the theoretical models to account for them?

Are there plans to extend the experiments to a broader range of datasets and architectures? This would help in understanding the generalizability of your findings.

Could you provide a detailed analysis of the resource efficiency of the proposed methods, especially in terms of training time, inference speed, and energy consumption?

How sensitive are the AP2 and AP3 properties to changes in hyperparameters? An analysis or discussion on this would be beneficial for practitioners aiming to implement these methods.

---

> ### Author Response · Authors · 2023-11-23
> **Response for Reviewer EYVP**
>
> Thank you very much for sharing your questions and concerns about the paper. Please find our responses below.
>
> **1. Could you provide direct empirical comparisons of the AP2 and AP3 properties against traditional pruning baselines? How do these new methods fare in terms of conventional metrics like accuracy, FLOPs reduction, and memory footprint?**
>
> Note that, the AP2 and AP3 metrics are not presented in this paper as newly proposed pruning approaches. While AP2 may resemble weight magnitude pruning, it is not employed here as a method to prune the network. Instead, these metrics are utilized to elucidate why the accuracy of the network does not significantly decrease when optimal pruning is applied.
>
> It's noteworthy that, in future research, based on our in-depth analysis in this paper we plan to introduce a novel pruning approach utilizing the AP3 metric. This new approach will be compared with existing techniques to assess its efficacy in network compression.
>
> **2. Have you tested the robustness of the pruned networks using AP2 and AP3 against adversarial attacks? Additionally, how do these pruned networks perform under distribution shifts or in out-of-distribution scenarios?**
>
> Thank you for your inquiry. It's important to clarify that AP3 is not designed as a pruning approach but rather serves as an explanatory metric for network pruning in this study. However, in future research, as we introduce a novel pruning approach based on AP3, we will thoroughly investigate its robustness.
>
> **3. You mentioned deviations from expected results based on Lemma 2. Could you delve deeper into the causes of these discrepancies and potentially adjust the theoretical models to account for them?**
>
> To clarify, as given in Lemma 2 the performance deviation $D$ is bounded by the function $g$ that depends on hyperparameters $\epsilon$, the maximum eigenvalue of Hessian at optimal point $\omega^*$, and constant $C$. However, in practice computing the explicit value of g is intractable. It is important to analyze the relationship between the parameters above and performance deviation. We have shown in the experiments such a relationship as how g is an increasing function of $\epsilon$.
>
> **4. Are there plans to extend the experiments to a broader range of datasets and architectures? This would help in understanding the generalizability of your findings.**
>
> We'd like to point out that in the Appendix, we have provided an extensive range of experiments including various architectures. In this paper, we intend to provide an in-depth theoretical analysis of our pruning explanations along with evaluating them on common datasets CIFAR 10 and 100 and VGG, ResNet, and AlexNet networks. To our knowledge, we are the first to explain DNN pruning analytically from a novel information theoretical angle.
>
> **5. Could you provide a detailed analysis of the resource efficiency of the proposed methods, especially in terms of training time, inference speed, and energy consumption?**
>
> Thank you for the valid point but as mentioned in previous responses, no specific pruning technique is proposed in this paper.
>
> **6. How sensitive are the AP2 and AP3 properties to changes in hyperparameters? An analysis or discussion on this would be beneficial for practitioners aiming to implement these methods.**
>
> Thank you for your question. I appreciate the opportunity to provide more details. AP2 involves no hyperparameters; it is simply the norm of the difference of weight vectors. However, AP3, as outlined on page 8 following Figure 3, does have hyperparameters.
>
> To elaborate further, when estimating the KL divergence between two multivariate T-student distributions using Monte Carlo estimation, we encounter hyperparameters such as the number of groups, the number of samples, and the degree of freedom. Since there is no exact value for the KL divergence of two multivariate T-student distributions, we opted for an iterative approach with 100 iterations.
>
> Throughout this process, we systematically explore different values for each hyperparameter,  selecting those that lead to convergence in the estimation. This hyperparameter tuning is crucial for enhancing the reliability and effectiveness of the AP3 metric in our analyses.

---

### Official Review · Reviewer_4kq3 · 2023-11-01

**Soundness:** 1 poor
**Presentation:** 1 poor
**Contribution:** 2 fair
**Rating:** 3
**Confidence:** 2

**Summary:**

This paper attempts to interpret network pruning by employing a probabilistic latent space of DNN weights. Two projective patterns, AP2 and AP3, are introduced in elucidating the sparsity of weight metrices. Experiments with AlexNet, VGG16 and ResNet50 on Cifar10/100 validate the theoretical results of AP2 and AP3.

**Strengths:**

A theoretical explanation of model compression is a less explored research area. Research in this area would be beneficial to develop new model compress methods.

**Weaknesses:**

The major issue of the paper is writing. Frankly, I can’t follow the argument since the problem formulation. More details are provide in the Questions section below.

**Questions:**

Is there any justification of Eq. (1)? Why can the loss be represented as a correlation between the weighted combination of the networks and the label? What is the definition of $w_F$, etc.?

How $w^{(l)} \in R^{M^{l-1}\times M^l}$ is mapped by $P: R^{n\times M^l}$ to $R^n$?

$\tilde{w}^* = m^* \odot w^*$? To my understanding, the optimal sparse $\tilde{w}^*$ can be different to masked version of $w^*$. As the sparsity pattern changes, the optimal weights can be very different to optimal weights of a full model.

---

> ### Author Response · Authors · 2023-11-22
> **Response for Reviewer 4kq3**
>
> Thank you very much for taking the time to provide us with your questions and concerns regarding the paper. Please see our responses below.
>
> **1. Is there any justification of Eq. (1)? Why can the loss be represented as a correlation between the weighted combination of the networks and the label? What is the definition of $w_{F}$, etc.?**
>
> Eq. (1) provides one formulation for defining the loss function, and similar expressions can be found in the following papers. In the equation, $w_{F}$ denotes the weight vector of network $F$. For specific details on the loss function, please refer to Equations 3 and 2 in the referenced papers, respectively.
>
> Ilyas A, Santurkar S, Tsipras D, Engstrom L, Tran B, Madry A. Adversarial examples are not bugs, they are features. Advances in neural information processing systems, 2019.
>
> Craig J, Andle J, Nowak TS, Sekeh SY. A Theoretical Perspective on Subnetwork Contributions to Adversarial Robustness. arXiv preprint arXiv:2307.03803, 2023.
>
> **2. How $\omega^{(l)}\in \mathbb{R}^{M_{l-1}\times M_l}$
>  is mapped by
> $\mathbb{R}^{n\times M_l}$
>  to
>  $\mathbb{R}^{n}$
> ?**
>
> Thank you for bringing this to our attention. The typo is fixed in the revision.
>
> **3. ${\widetilde{\omega}}^\*=m^\*\odot \omega^\*$? To my understanding, the optimal sparse ${\widetilde{\omega}}^\*$
>  can be different to masked version of $\omega^\*$. As the sparsity pattern changes, the optimal weights can be very different from the optimal weights of a full model.**
>
>  Indeed,  in our paper, the notation $m^*$  represents the optimal mask, signifying the most effective mask among all possible sparsity patterns. Hence, we refer to
>  $\widetilde{\omega}^*$ as the optimal masked version of  $\omega^*$. Throughout the paper, the use of $*$ consistently denotes the optimal version.

---

### Official Review · Reviewer_amvG · 2023-11-02

**Soundness:** 1 poor
**Presentation:** 1 poor
**Contribution:** 2 fair
**Rating:** 3
**Confidence:** 3

**Summary:**

The authors investigate the correlations between parameter compression error and performance difference in neural networks and their pruned counterparts. The compression error is computed either through the squared Frobenius norm of the parameters' projection to another space (though the authors generally use identity projection), or mapping the parameters to a space of probability measures and computing the KL divergence.

**Strengths:**

- The authors' choice of topic is a crucial one: Understanding why and when compression works is important not only for better compression methodology but for extracting insights with learning theoretical implications.

**Weaknesses:**

- The paper fails to make a distinguishable contribution to understanding why and when compression works. It does not live up to the term "explaining" in its title since it contains no arguments why or when compression should be applicable without harming performance. In my estimation the paper ends up correlating some probabilistic divergences with performance difference between original and pruned network, but even then 1- using the Frobenius norm of the parameters' difference seems to work better, 2- the mapping of parameters to probabilistic measures are not motivated in any systematic way.
- The paper is inconsistent with its terminology and notation, and omits some fundamental information or concepts when making its case. I provide a more detailed, chronological list of the points that I found problematic.

**Questions:**

Here I present some questions and observations from the text that I found difficult to understand:
- Pg. 1 onwards: I find the usage of "probability space" and related terms confusing throughout the paper. Here $\mathcal{Z}$ is introduced as a "probability space" but I strongly suspect the authors mean the domain (state space) of the random variable $Z$. Similar for $\tilde{Z}$.
- Pg. 1: Neither in the beginning, nor anywhere the paper makes an explicit, mathematical definition of compression. Given the centrality of the notion for the paper, and the various ways in which it can be defined, I find this a very important shortcoming.
- Pg. 2: What is a "probabilistic space"? If the intended meaning is probability space, more care should be taken for using the established terminology. If not, explicit definition is required.
- Pg. 2: I find it extremely confusing that the authors use the same terminology for the parameter matrix and its vectorized version. This confusion goes beyond this point as well. E.g. although the authors state that they will proceed with the vectorized parameters, Definition 1 clearly refers to parameter matrix.
- Pg. 2: In (1), why do the authors refer to a weighted combination of models in the model space? As far as I can see this construction never referred to again.
- Pg. 2: What are the domains of $\mathbf{X}$ and $Y$?
- Pg. 2: I think the paper needs to justify that $\tilde{w}^* = m^* \odot w^*$, that the ideal compressed parameters are a masked version of the ideal uncompressed parameters. But given no explicit definition of compression/pruning has been presented, this is difficult to do here.
- Pg. 2 Definition 2: What is $n$? Above the authors used $\mathbb{R}^{M_{l-1}\times M_l}$, why use a different notation now?
- Pg. 2 Definition 2: What is a filter? Above $f^{(l)}$ has been explicitly introduced as a network layer. But in this definition $f^{(l)}$ is called a "filter" and the layers are referred to through scalar $l$. I think that for a paper with theoretical aspirations the notations and terminology are too chaotic.
- Pg. 2: The paper refers to $\omega^{(l)}$ as both vectors and "parameter set"s. What is a parameter set? Why is the naming for this term changing?
- Pg. 2: "$\mathcal{Z}$ and $\tilde{\mathcal{Z}}$ are latent spaces that could overlap." Should these two spaces not be the same in order for KL divergence to be computable? What are we integrating over in Eq. 2, if not?
- Pg. 4 Definition 5: Why is the probability space in the first sentence defined? Is it used in any way? Is $\mathcal{F}$ the event space here, was this notation not reserved for the model space? I strongly suspect that the authors intended mapping is to a space of probability spaces defined on the same measurable space, with the state space being $\mathcal{Z} = \tilde{\mathcal{Z}}$. If this is true, the definition strays far away from this purpose, and the use of terms such as "probabilistic space" adds to the confusion.
- Pg. 5 Theorem 1: The theorem starts with a compressed training scheme. This is not introduced beforehand or afterwards.
- Pg. 6: How are these models pre-trained?
- Pg. 8: Given these results, do not the papers' findings boil down to suggesting using Frobenius norm of the original and pruned parmaeters difference for examining compression and performance? If so, is this not a very novel finding, and should be considered in light of previous results such as [1].

[1] Y. Jiang*, B. Neyshabur*, H. Mobahi, D. Krishnan, and S. Bengio, “Fantastic Generalization Measures and Where to Find Them,” presented at the International Conference on Learning Representations, Sep. 2019. Accessed: Aug. 05, 2020. [Online]. Available: https://openreview.net/forum?id=SJgIPJBFvH

---

### Note after Rebuttals

I thank the authors for their feedback, which clarified a number of confusing points in the paper, as well as some details I missed. Unfortunately, my fundamental concerns regarding the contributions of the paper as well as their presentation still mostly remain, based on which I retain my recommendation.

---

> ### Author Response · Authors · 2023-11-23
> **Response for Reviewer amvG**
>
> Thank you for providing your concerns and thoughts on the work so that we can work to revise and improve it. We've addressed and we hope that our revisions and responses found below help clarify our work more.
>
> **Pg. 1 onwards**
>
> We recognize potential confusion and agree with your suggestion to clarify. Regarding $\mathcal {Z}$, our intent is to represent a probability space, denoted as $(\Omega, \mathcal{F}, P)$.
>
> **Pg. 1**
>
>  As stated in the abstract, our focus in this paper is on pruning as a compression method. We provide the mathematical definition of pruning on page 2, immediately following Equation 1. "Note ${\widetilde{\omega}}^*=m^*\odot \omega^*$ is optimal sparse weight where $m\in[0,1]^d$ is the binary mask matrix ($d$ is dimension). The optimal sparse weight is achieved by  $(m^*, \omega^*):= \underset{\omega,m}{\arg \min}  \mathbb{E}_{(\mathbf{X},Y)\sim D} \\{ Y \cdot \ell(\omega) \\}$".
> This definition aligns with standard approaches, and we refer to a specific paper as an example.
>
>  Blalock, Davis and Gonzalez Ortiz, Jose Javier and Frankle, Jonathan, and Guttag, John, "What is the State of Neural Network Pruning?," Proceedings of Machine Learning and Systems, 2020 (2), 129--146.
>
> **Pg. 2: What is a probabilistic space?**
>  By probabilistic space, we meant the same as probability space and we fixed this in the revised version.
>
> **Pg. 2: I find it confusing**
>
> We maintain consistent notation for both weight matrices and vectors for simplicity, explicitly noting in the Notations section that weight tensors can be viewed as high-dimensional vectors. We have made adjustments in the introduction for enhanced clarity and acknowledge and correct the noted typo in Definition 1.
>
> **Pg. 2: In (1)**
>
> Indeed, it is a common definition of loss function.
>  Moreover, we defined $\ell(\omega)$ as the weighted combination of the networks and we used this definition in our proofs.
> For reference, Please see the loss function definitions in equations 3 and 2 in the following papers, respectively.
>
> 1. Ilyas A, Santurkar S, Tsipras D, Engstrom L, Tran B, Madry A. Adversarial examples are not bugs, they are features. Advances in neural information processing systems (NeurIPs), 2019.
>
> 2. Craig J, Andle J, Nowak TS, Sekeh SY. A Theoretical Perspective on Subnetwork Contributions to Adversarial Robustness. arXiv preprint arXiv:2307.03803, 2023.
>
> **Pg. 2: What are the domains**
> The input $\mathbf{X}\in \mathcal{X}$ and target $Y\in \mathcal{Y}$ have realization space $\mathcal{X}$ and $\mathcal{Y}$, respectively with joint distribution $D$. We have restated this in the revision to clarify this.
>
> **Pg. 2: I think**
>
> As previously mentioned, employing  ${\widetilde{\omega}}^*=m^*\odot \omega^*$ serves as one approach to define pruning.
>
> **Pg. 2 Definition 2: What is $n$?**
>
> You are correct. This was a typo and is fixed in the revision.
>
> **Pg. 2 Definition 2: What is a filter?**
>
> In general, the notation $f^{(l)}$ (referring to the l-th layer of the network) denotes the filters of that particular layer. We have clarified this in the revised paper.
>
> **Pg. 2: The paper**
>
> The term "parameter set" refers to the parameters associated with a distribution. The name of the term $\omega^{(l)}$ remains unchanged but now serves as a parameter of the distribution. The revised sentence in the updated version aims to provide clarity.
>
> **Pg. 2: $\mathcal{Z}$ and $\mathcal{\widetilde{Z}}$**
>
> To have a compatible kl divergence between two distributions, they should have the same domain. So either two probability spaces should have some common elements or be aligned. Although, the distributions can be different or have different parameters. The integration is on the same domain of distributions.
>
> **Pg. 4**
>
> Thank you for pointing that out. We acknowledge the error, and it has been appropriately addressed in the revised version.
>
> **Pg. 5**
>
> Thank you for the feedback. We have rephrased the expression to convey the concept in an alternative manner.
>
> **Pg. 6**
>
> As indicated in the "EXPERIMENTS" section, the models utilized in our study have been pre-trained on the Imagenet dataset. We employed pre-trained models available in the PyTorch framework.
>
> **Pg. 8**
>
> Our paper primarily offers a theoretical explanation of deep neural network pruning. Our objective is not to advocate for a specific pruning approach or demonstrate the superiority of our method over others. Rather, we provide an explanation of the pruning and performance relationship using the Frobenius norm. It's important to note that our experiments serve as evidence supporting our theorems, illustrating why optimal pruning leads to nearly identical accuracy. However, in future works, we plan to propose a novel pruning approach that leverages the AP3 metric and our studies in this paper. This involves fine-tuning hyperparameters and investigating various non-Gaussian distributions to achieve improved accuracy for the pruned network, compared to utilizing the Frobenius norm.

---

### Meta-Review · Area_Chair_dwkd · 2023-12-06

**Metareview:**

The paper presents an analysis of compression and relation pruning in neural networks. The merit of the paper lies in its theoretical framework of compression, a seldom explored but very important research area. However, despite consensus among reviewers on the conceptual importance, the writing, presentation and explanations in the paper presently hold it back from living up to its title's expectation. Among some of the questions that may have been addressed in a rebuttal, main points for prospective improvements remain with respect to clarifying the precise distinguishable contribution, disambiguating notation and partially confusing structure, as well as justifying some of the assumptions/components.

**Justification For Why Not Higher Score:**

Many reviewers raised a valid amount of concerns, noted down questions with respect to parts of the paper that remain unclear, expressed concerns about the quality of the writing, and most importantly, as the shared main weaknesses, requested clarifications with respect to a distinguishable contribution and justification of employed methodology. Whereas some of the more nuanced questions, e.g. on notation and correctness have been addressed in a rebuttal, the latter main concerns do not seem to have been addressed, neither in a rebuttal nor in a pdf revision.

**Justification For Why Not Lower Score:**

N/A

---

### Decision · Program_Chairs · 2024-01-16

Reject